# Competitive binding of STATs to receptor phospho-Tyr motifs accounts for altered cytokine responses

Stephan Wilmes[1†], Polly-Anne Jeffrey[2†], Jonathan Martinez-Fabregas[1], Maximillian Hafer[3], Paul K Fyfe[1], Elizabeth Pohler[1], Silvia Gaggero[4], Martín López-García[2], Grant Lythe[2], Charles Taylor[5], Thomas Guerrier[6], David Launay[6], Suman Mitra[4], Jacob Piehler[3], Carmen Molina-París[2,7]*, Ignacio Moraga[1]*

[1]Division of Cell Signalling and Immunology, School of Life Sciences, University of Dundee, Dundee, United Kingdom; [2]Department of Applied Mathematics, School of Mathematics, University of Leeds, Leeds, United Kingdom; [3]Department of Biology and Centre of Cellular Nanoanalytics, University of Osnabrück, Osnabrück, Germany; [4]Université de Lille, INSERM UMR1277 CNRS UMR9020–CANTHER and Institut pour la Recherche sur le Cancer de Lille (IRCL), Lille, France; [5]Department of Statistics, School of Mathematics, University of Leeds, Leeds, United Kingdom; [6]Univ. Lille, Univ. LilleInserm, CHU Lille, U1286 - INFINITE - Institute for Translational Research in Inflammation, Lille, France; [7]T-6 Theoretical Division, Los Alamos National Laboratory, Los Alamos, United States

*For correspondence:
C.MolinaParis@leeds.ac.uk (CM-P); imoragagonzalez@dundee.ac.uk (IM)

†These authors contributed equally to this work

Competing interests: The authors declare that no competing interests exist.

**Abstract** Cytokines elicit pleiotropic and non-redundant activities despite strong overlap in their usage of receptors, JAKs and STATs molecules. We use IL-6 and IL-27 to ask how two cytokines activating the same signaling pathway have different biological roles. We found that IL-27 induces more sustained STAT1 phosphorylation than IL-6, with the two cytokines inducing comparable levels of STAT3 phosphorylation. Mathematical and statistical modeling of IL-6 and IL-27 signaling identified STAT3 binding to GP130, and STAT1 binding to IL-27Rα, as the main dynamical processes contributing to sustained pSTAT1 levels by IL-27. Mutation of Tyr613 on IL-27Rα decreased IL-27-induced STAT1 phosphorylation by 80% but had limited effect on STAT3 phosphorgylation. Strong receptor/STAT coupling by IL-27 initiated a unique gene expression program, which required sustained STAT1 phosphorylation and IRF1 expression and was enriched in classical Interferon Stimulated Genes. Interestingly, the STAT/receptor coupling exhibited by IL-6/IL-27 was altered in patients with systemic lupus erythematosus (SLE). IL-6/IL-27 induced a more potent STAT1 activation in SLE patients than in healthy controls, which correlated with higher STAT1 expression in these patients. Partial inhibition of JAK activation by sub-saturating doses of Tofacitinib specifically lowered the levels of STAT1 activation by IL-6. Our data show that receptor and STATs concentrations critically contribute to shape cytokine responses and generate functional pleiotropy in health and disease.

## Introduction

IL-27 and IL-6 both have intricate functions regulating inflammatory responses (*O'Shea and Plenge, 2012*). IL-27 is a hetero-dimeric cytokine comprised of p28 and EBI3 subunits (*Pflanz et al., 2002*). IL-27 exerts its activities by binding GP130 and IL-27Rα receptor subunits in the surface of responsive cells, triggering the activation of the JAK1/STAT1/STAT3 signaling pathway. IL-27 elicits both

pro- and anti-inflammatory responses, although the later activity seems to be the dominant one (*Yoshida and Hunter, 2015*). IL-27 stimulation inhibits RORγt expression, thereby suppressing Th-17 commitment and limiting subsequent production of pro-inflammatory IL-17 (*Stumhofer et al., 2006*; *Diveu et al., 2009*). Moreover, IL-27 induces a strong production of anti-inflammatory IL-10 on (Tbet$^+$ and FoxP3$^-$) Tr-1 cells (*Fitzgerald et al., 2007*; *Stumhofer et al., 2007*; *Pot et al., 2011*) further contributing to limit the inflammatory response. IL-6 engages a hexameric receptor complex comprised each of two copies of IL-6Rα, GP130 and IL-6 (*Boulanger et al., 2003*), triggering the activation, as IL-27 does, of the JAK1/STAT1/STAT3 signaling pathway. However, opposite to IL-27, IL-6 is known as a paradigm pro-inflammatory cytokine (*Rose-John, 2018*; *Hunter and Jones, 2015*). IL-6 inhibits lineage differentiation to Treg cells (*Korn et al., 2008*) while promoting that of Th-17 cells (*Kimura and Kishimoto, 2010*; *Jones et al., 2010*), thus supporting its pro-inflammatory role. How IL-27 and IL-6 elicit opposite immunomodulatory activities despite activating almost identical signaling pathways is currently not completely understood.

The relative and absolute STAT activation levels seem to have a diverse set of roles, which lead to a strong signaling and functional plasticity by cytokines. Although IL-6 robustly activates STAT3, it is capable to mount a considerable STAT1 response as well (*Rolvering et al., 2017*). Moreover, in the absence of STAT3, IL-6 induces a strong STAT1 response comparable to IFNγ – a prototypic STAT1 activating cytokine (*Costa-Pereira et al., 2002*). Likewise, the absence of STAT1 potentiates the STAT3 response for IL-27, which normally elicits a strong STAT1 response, rendering it to mount an IL-6-like response (*Rolvering et al., 2017*). This suggests a competition of STAT1/3 for phospho-tyrosine motifs at the cytoplasmic domain of cytokine receptors regulated by their different binding rate constants (on and off rates). Indeed, different STAT1 or STAT3 binding affinities have been assessed in vitro for the phospho-tyrosines on GP130 (*Wiederkehr-Adam et al., 2003*). Furthermore, negative feedback mechanisms, controlled by SOCSs and phosphatases, have been described as critical players influencing STAT1 and STAT3 phosphorylation kinetics and thereby shaping their signal integration for GP130-utilizing cytokines (*Schmitz et al., 2000*; *Yasukawa et al., 2003*; *Croker et al., 2003*; *Brender et al., 2007*). Yet, how all these molecular components are integrated by a given cell to produce the desired response is still an open question. Among the IL-6/IL-12 cytokine family, IL-27 exhibits a unique STAT activation pattern. The majority of GP130-engaging cytokines activate preferentially STAT3, with activation of STAT1 an accessory or balancing component in the signaling pathway (*Camporeale, 2012*; *Regis et al., 2008*). IL-27, however, triggers STAT1 and STAT3 activation with high potency (*Lucas et al., 2003*). Indeed, different studies have shown that IL-27 responses rely on either STAT1 (*Kamiya et al., 2004*; *Takeda et al., 2003*; *Neufert et al., 2007*) or STAT3 activation (*Stumhofer et al., 2007*; *Owaki et al., 2008*). Moreover, recent transcriptomics studies showed that in the absence of STAT3, IL-6 and IL-27 lost more than 75% of target gene induction. Yet, STAT1 was the main factor driving the specificity of the IL-27 versus the IL-6 response, highlighting a critical interplay of STAT1 and STAT3 engagement (*Hirahara et al., 2015*).

While the biological responses induced by IL-27 and IL-6 have been extensively studied (*Yoshida and Hunter, 2015*; *Hunter and Jones, 2015*), the very initial steps of signal activation and kinetic integration by these two cytokines have not been comprehensively analysed. Since the different biological outcomes elicited by IL-27 and IL-6 are most likely encoded in the early events of cytokine stimulation, here we specifically aimed to identify the molecular determinants underlying functional selectivity by IL-27 in human T-cells. We asked how a defined cytokine stimulus is propagated in time over multiple layers of signaling to produce the desired response. To this end, we probed IL-27 and IL-6 signaling at different scales, ranging from cell surface receptor assembly and early STAT1/3 effector activation to an unbiased and quantitative multi-omics approach: phospho-proteomics after early cytokine stimulation, kinetics of transcriptomic changes and alteration of the T-cell proteome upon prolonged cytokine exposure.

IL-6 and IL-27 induced similar levels of assembly of their respective receptor complexes, which resulted in comparable phosphorylation of STAT3 by the two cytokines. IL-27, on the other hand, triggered a more sustained STAT1 phosphorylation. To decipher the molecular events which determine sustained STAT1 phosphorylation by IL-27, we mathematically model the STAT1 and STAT3 signaling kinetics induced by each of these cytokines. We identified differential binding of STAT1 and STAT3 to IL-27Rα and GP130, respectively, as the main factors contributing to a sustained STAT1 activation by IL-27. At the transcriptional level, IL-27 triggered the expression of a unique gene program, which strictly required the cooperative action between sustained pSTAT1 and IRF1

expression to drive the induction of an interferon-like gene signature that profoundly shaped the T-cell proteome. Interestingly, our mathematical models of IL-6 and IL-27 signaling predicted that changes in receptor and STAT expression could fundamentally change the magnitude and timescale of the IL-6 and IL-27 responses. We found high levels of STAT1 expression in SLE patients when compared to healthy donors, which correlated with biased STAT1 responses induced by IL-6 and IL-27 in these patients. Strikingly, we could specifically inhibit STAT1 activation by IL-6 using suboptimal doses of the JAK inhibitor Tofacitinib. This could provide a new strategy to specifically target individual STATs engaged by cytokines.

## Results

### IL-27 induces a more sustained STAT1 activation than HypIL-6 in human Th-1 cells

IL-6 and IL-27 are critical immunomodulatory cytokines. Although IL-6 engages a hexameric surface receptor comprised of two molecules of IL-6Rα and two molecules of GP130 to trigger the activation of STAT1 and STAT3 transcription factors (*Figure 1a*), IL-27 binds GP130 and IL-27Rα to trigger activation of the same STATs molecules (*Figure 1a*). Despite sharing a common receptor subunit, GP130, and activating similar signaling pathways, these two cytokines exhibit non-redundant immunomodulatory activities, with IL-6 eliciting a potent pro-inflammatory response and IL-27 acting more as an anti-inflammatory cytokine. Here, we set to investigate the molecular rules that determine the functional specificity elicited by IL-6 and IL-27 using human Th-1 cells as a model experimental system. Due to the challenging recombinant expression of the human IL-27, we have recombinantly produced a murine single-chain variant of IL-27 (p28 and EBI3) which cross-reacts with the human receptors and triggers potent signaling, comparable to the signaling output produced by commercial human IL-27 (*Oniki et al., 2006*; *Figure 1—figure supplement 1a*). In addition, we have used a linker-connected single-chain fusion protein of IL-6Rα and IL-6 termed HyperIL-6 (HypIL-6) (*Fischer et al., 1997*) to diminish IL-6 signaling variability due to changes in IL-6Rα expression during T cell activation (*Oberg et al., 2006*).

CD4 +T cells from human buffy coat samples were isolated by magnetic activated cell sorting (MACS) and grew under Th-1-polarizing conditions. Th-1 cells were then used to study in vitro signaling by IL-27 and IL-6 (*Figure 1—figure supplement 1b*). We took advantage of a barcoding methodology allowing high-throughput multiparameter flow cytometry to perform detailed dose/response and kinetics studies induced by HypIL-6 and IL-27 in Th-1 cells (*Krutzik et al., 2011*; *Figure 1—figure supplement 1b*). Dose-response experiments with IL-27 and HypIL-6 on Th-1 cells showed concentration-dependent phosphorylation of STAT1 and STAT3. Phosphorylation of STAT1/3 was more sensitive to activation by IL-27 with an $EC_{50}$ of ~20 pM compared to ~400 pM for HypIL-6 (*Figure 1b*). Despite this difference in sensitivity, both cytokines yielded the same activation amplitude for pSTAT3. For pSTAT1, however, we observed a significantly reduced maximal amplitude for HypIL-6 relative to IL-27 (*Figure 1b*). We next performed kinetic studies to assess whether the poor STAT1 activation by HypIL-6 was a result from different activation kinetics. For STAT3, we saw the peak of phosphorylation after ~15–30 min, followed by a gradual decline. Both cytokines exhibited an almost identical sustained pSTAT3 profile, with ~20% of activation still seen after 3 hr of continuous stimulation. Interestingly, IL-27 did not only activate STAT1 with higher amplitude but also more sustained than HypIL-6 (*Figure 1c*). This could be better appreciated when pSTAT1 levels were normalized to maximal MFI for each cytokine, with IL-27 inducing clearly a more sustain phosphorylation of STAT1 than HypIL-6 (*Figure 1—figure supplement 1c*). The same phenotype was observed in other T-cell subsets of activated PBMCs (*Figure 1—figure supplement 1d*). As cell surface GP130 levels are significantly reduced upon T-cell activation (*Betz and Müller, 1998*), we next investigated whether the transient STAT1 activation profile induced by HypIL-6 resulted from limited availability of GP130. For that we generated a RPE1 cell clone stably expressing 10 times higher levels of GP130 in its surface (*Figure 1d*, right panel). Stimulation of this RPE1 clone with HypIL-6 resulted in a more sustained activation of STAT3, with very little effect on STAT1 activation kinetics when compared to RPE1 wild-type cells, suggesting that GP130 receptor density does not contribute to the transient STAT1 activation kinetics elicited by HypIL-6 (*Figure 1d*).

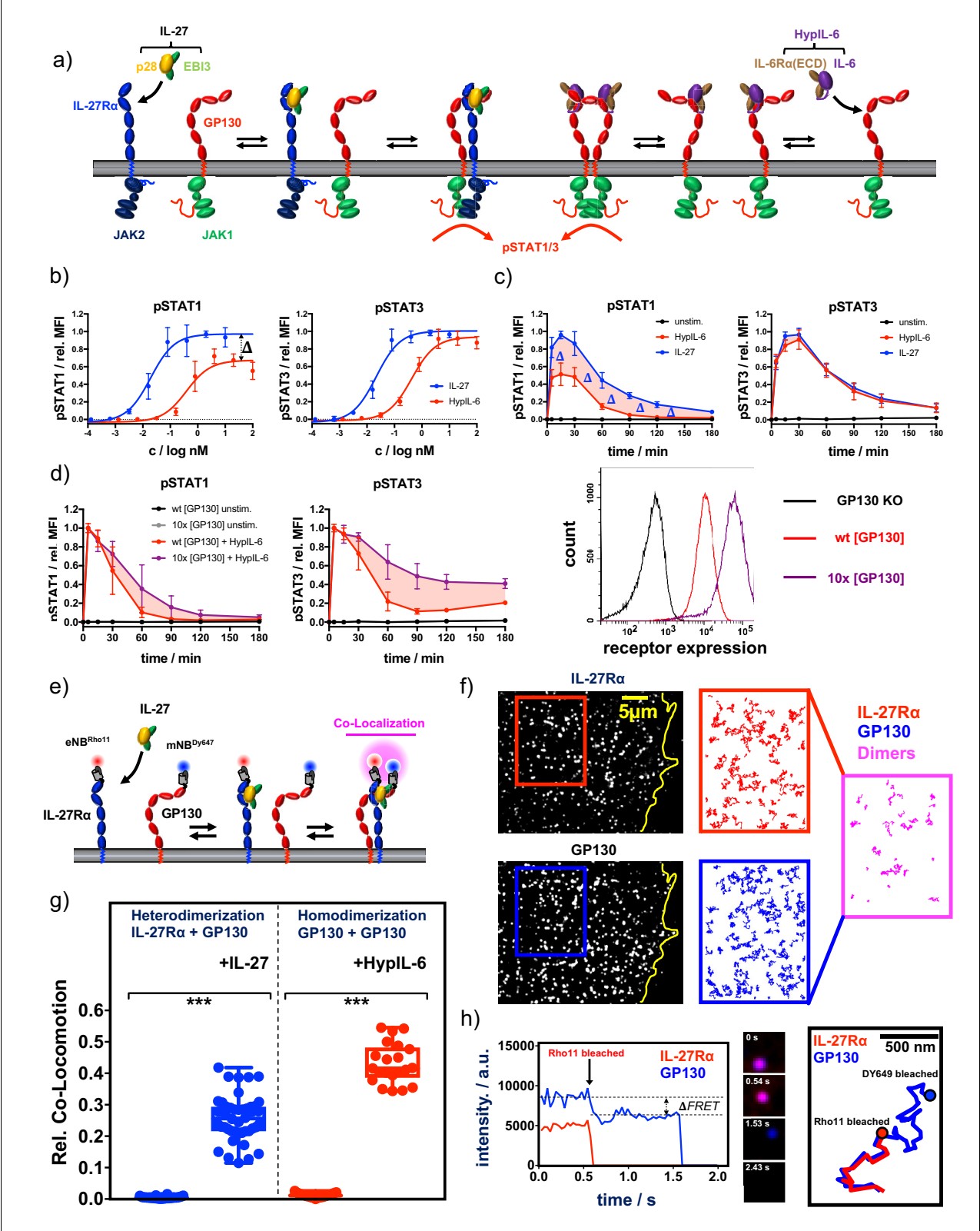

**Figure 1.** Cytokine receptor activation by IL-27 and (Hyp)IL-6: (**a**) Cartoon model of stepwise assembly of the IL-27 and HypIL-6-induced receptor complex and subsequent activation of STAT1 and STAT3. (**b**) Dose-dependent phosphorylation of STAT1 and STAT3 as a response to IL-27 and HypIL-6 stimulation in TH-1 cells, normalized to maximal IL-27 stimulation. Data was obtained from three biological replicates with each two technical replicates, showing mean ±std dev. (**c**) Phosphorylation kinetics of STAT1 and STAT3 followed after stimulation with saturating concentrations of IL-27 (2

*Figure 1 continued on next page*

*Figure 1 continued*

nM) and HypIL-6 (20 nM) or unstimulated TH-1 cells, normalized to maximal IL-27 stimulation. Data was obtained from five biological replicates with each two technical replicates, showing mean ±std dev. (**d**) Top: Phosphorylation kinetics of STAT1 and STAT3 followed after stimulation with HypIL-6 (20 nM) or left unstimulated, comparing wt RPE1 and RPE1 GP130KO reconstituted with high levels of mXFPm-GP130 (=10 x [GP130]). Data was normalized to maximal stimulation levels of each treatment. Left: cell surface GP130 levels comparing RPE1 GP130KO, wt RPE1 and RPE1 GP130KO stably expressing mXFPm-GP130 measured by flow cytometry. Data was obtained from one biological replicate with each two technical replicates, showing mean ±std dev. Bottom right: cell surface levels of GP130 measured by flow cytometry for indicated cell lines. (**e**) Cartoon model of cell surface labeling of mXFP-tagged receptors by dye-conjugated anti-GFP nanobodies (NB) and identification of receptor dimers by single molecule dual-color co-localization. (**f**) Raw data of dual-color single-molecule TIRF imaging of mXFPe-IL-27Rα$^{NB-RHO11}$ and GP130$^{NB-DY649}$ after stimulation with IL-27. Particles from the insets (IL-27Ra: red and GP130: blue) were followed by single molecule tracking (150 frames ~ 4.8 s) and trajectories > 10 steps (320 ms) are displayed. Receptor heterodimerization was detected by co-localization/co-tracking analysis. (**g**) Relative number of co-trajectories observed for heterodimerization of IL-27Rα and GP130 as well as homodimerization of GP130 for unstimulated cells or after indicated cytokine stimulation. Each data point represents the analysis from one cell with a minimum of 23 cells measured for each condition. Two-tailed Student's T-test: *p<0.05, **p≤0.01, ***p≤0.001; n.s., not significant. (**h**) Stoichiometry of the IL-27–induced receptor complex revealed by bleaching analysis. Left: Intensity traces of mXFPe-IL-27Rα$^{NB-RHO11}$ and GP130$^{NB-DY649}$ were followed until fluorophore bleaching. Middle: Merged imaging raw data for selected timepoints. Right: overlay of the trajectories for IL-27Rα (red) and GP130 (blue).

The online version of this article includes the following figure supplement(s) for figure 1:

**Figure supplement 1.** Characterization of IL-27 and HypIL-6 signaling in T-cell subsets followed by high-throughput flow-cytometry.
**Figure supplement 2.** Functional characterization of RPE1 IL-27Rα.
**Figure supplement 3.** Ligand-induced receptor assembly - stoichiometry and diffusion properties.

## Ligand-induced cell-surface receptor assembly by IL-27 and HypIL-6

We next investigated whether IL-27 and HypIL-6 elicited differential cell surface receptor engagement that could explain their distinct signaling output. For that, we measured the dynamics of receptor assembly in the plasma membrane of live cells by simultaneous dual-color total internal reflection fluorescence (TIRF) imaging. RPE1 cells were chosen as a model experimental system since they do not express endogenous IL-27Rα (*Figure 1—figure supplement 2a*). We generated an RPE1 clone, stably expressing IL-27Rα which responds to IL-27 and HypIL-6 stimulation. Importantly, this reconstituted experimental system mimicked the pSTAT1/3 activation kinetics of T-cells (Supp. Fig. 2b), albeit exhibiting similar pSTAT1 amplitudes for IL-27 and HypIL-6, most likely because of the high endogenous levels of GP130 in RPE1 cells. We used previously described RPE1 GP130 KO cells (*Figure 1—figure supplement 2*; *Martinez-Fabregas et al., 2019*) to transfect and express tagged variants of IL-27Rα and GP130, to allow quantitative site-specific fluorescence cell surface labeling by dye-conjugated nanobodies (NBs)

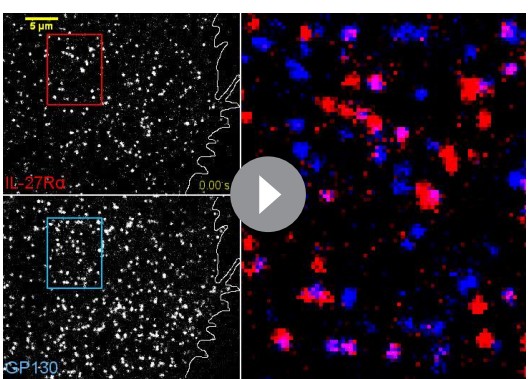

**Video 1.** Single-molecule co-tracking as a readout for dimerization of cytokine receptors. Cell surface labeling of mXFPe-IL-27Rα by eNB$^{RHO11}$ (left, top) and mXFPm-GP130 by mNB$^{DY649}$ (left, bottom) after stimulation with IL-27 (20 nM). In the overlay of the zoomed section of both spectral channels (mXFPe-IL-27Rα$^{RHO11}$: Red, mXFPm-GP130$^{DY649}$: Blue), yellow lines indicate co-locomotion of IL-27Rα and GP130 (≥10 steps). Acquisition frame rate: 30 Hz, Playback: real time.
https://elifesciences.org/articles/66014#video1

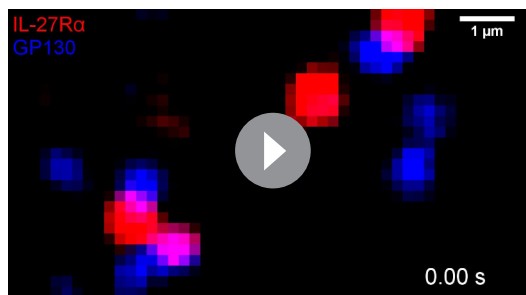

**Video 2.** Dynamics of IL-27-induced receptor assembly. Formation of a single-molecule heterodimer of mXFPe-IL-27Rα$^{RHO11}$ (Red) and mXFPm-GP130$^{DY649}$ (Blue) in presence of IL-27. Yellow lines indicate co-locomotion of IL-27Rα and GP130 (≥10 steps). Acquisition frame rate: 30 Hz, Playback: real time with break at time of receptor dimerization.
https://elifesciences.org/articles/66014#video2

(*Figure 1e*) as recently described in *Gorby et al., 2020*. For both IL-27Rα and GP130, we found a random distribution and unhindered lateral diffusion of individual receptor monomers (*Figure 1f*). Single-molecule co-localization combined with co-tracking analysis was then used to identify correlated motion of IL-27Rα and GP130 which was taken as a readout for receptor heterodimer formation (*Ruprecht et al., 2010*; *Figure 1f*, *Video 1*). In the resting state, we did not observe pre-assembly of IL-27Rα and GP130. However, after stimulation with IL-27 we found substantial heterodimerization (*Figure 1f and g*, *Figure 1—figure supplement 3a*, *Video 1* and *2*). At elevated laser intensities, bleaching analysis of individual complexes confirmed a one-to-one (1:1) complex stoichiometry of IL-27Rα and GP130, whereas single-molecule Förster resonance energy transfer (FRET) further corroborated close molecular proximity of the two receptor chains (*Figure 1h*). We also observed association and dissociation events of receptor heterodimers, pointing to a dynamic equilibrium between monomers and dimers as proposed for other heterodimeric cytokine receptor systems (*Moraga et al., 2015a*; *Wilmes et al., 2015*; *Video 3*).

To measure homodimerization of GP130 by HypIL-6, we stochastically labeled GP130 with equal concentrations of the same NB species conjugated to either of the two dyes (*Wilmes et al., 2020*). We saw strong homodimerization of GP130 after stimulation with HypIL-6 (*Figure 1g*, *Figure 1—figure supplement 3a*, *Video 4*). Homodimerization was confirmed either by single-color dual-step bleaching or dual-color single-step bleaching as shown for other homodimeric cytokine receptors (*Figure 1—figure supplement 3b*; *Pflanz et al., 2004*). For both cytokine receptor systems, we saw a cytokine-induced reduction of the diffusion mobility, which has been ascribed to increased friction of receptor dimers diffusing in the plasma membrane. However, we note that HypIL-6 stimulation impaired diffusion of GP130 more strongly than IL-27 did, possibly indicating faster receptor internalization (*Figure 1—figure supplement 3c*). Based on the dimerization data, we were able to calculate the two-dimensional equilibrium dissociation constants ($K_D^{2D}$) assuming mass action kinetics for a dynamic monomer-dimer equilibrium: for IL-27-induced heterodimerization of IL-27Rα and GP130, we calculated a two-dimensional constant, denoted by $K_D^{2D}$, of approximately 0.81 μm$^{-2}$. In activated T-cells with high levels and a significant excess of IL-27Rα over GP130, this $K_D^{2D}$ ensures strong receptor assembly by IL-27 (*Diegelmann et al., 2012*). $K_D^{2D}$ for GP130 homodimerization by HypIL-6 was approximately 0.21 μm$^{-2}$. This higher affinity is most likely due to the two high-affinity binding sites engaged in the hexameric receptor complex (*Boulanger et al., 2003*). However, in T-cells the expression of GP130 can be particularly low, thus, probably limiting HypIL-6-induced signaling. Taken together, these experiments marked ligand-induced receptor assembly as the initial step triggering downstream signaling for both IL-27 and HypIL-6, with no obvious differences in their receptor activation mechanism which could support the observed more sustained STAT1 activation elicited by IL-27.

## Mathematical and statistical analysis of HypIL-6 and IL-27-induced STAT kinetic responses

To gain further insight into the molecular rules and kinetics that define IL-27 sustained STAT1 phosphorylation, we developed two mathematical models of the initial steps of HypIL-6 and IL-27 receptor-mediated signaling, respectively. A diagram which describes the molecular reactions in each model is shown in *Figure 2—figure supplement 1a*, and the complete model reaction scheme is given in *Figure 2—figure supplement 1b–h*. The mathematical model for each cytokine considers the following events: (i) cytokine association and dissociation to a receptor chain (*Figure 2a*, *Figure 2—figure supplement 1b and c*, top panel), (ii) cytokine-induced dimer association and dissociation (*Figure 2—figure supplement 1b and c*, bottom panel), (iii) STAT1

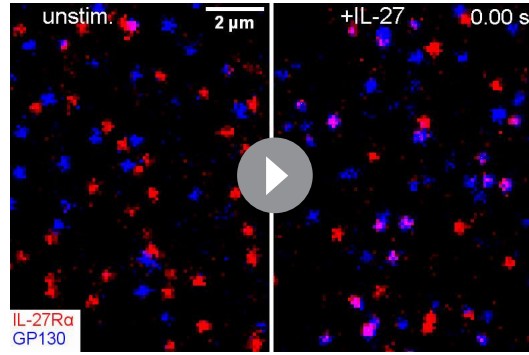

**Video 3.** Ligand-induced heterodimerization of IL-27Rα and GP130. Overlay of the two spectral channels (mXFPe-IL-27Rα$^{RHO11}$: Red, mXFPm-GP130$^{DY649}$: Blue) in absence (left) or presence (right) of IL-27 (20 nM). Yellow lines indicate co-locomotion of IL-27Rα and GP130 (≥10 steps). Acquisition frame rate: 30 Hz, Playback: real time.
https://elifesciences.org/articles/66014#video3

(or STAT3) binding and unbinding to dimer (*Figure 2—figure supplement 1d and e*), (iv) STAT1 (or STAT3) phosphorylation when bound to dimer (*Figure 2—figure supplement 1d and e*), (v) internalization/degradation of complexes (*Figure 2—figure supplement 1f and g*), and (vi) dephosphorylation of free STAT1 (or STAT3) (*Figure 2—figure supplement 1h*). Details of model assumptions, model parameters and parameter inference have been provided in the Materials and methods under the sections Mathematical models and Bayesian inference. The ordinary differential equations for each mathematical model are given in Supplementary Information 1, where we have assumed mass action kinetics.

We first wanted to explore if there existed a potential feedback mechanism in the mechanisms of receptor complexes internalization/degradation over time. Negative feedback mechanisms have been reported for cytokine receptor signaling, while positive feedback loops are unlikely to happen at the timescale of the given experiments (3 hr). To this end, and for each cytokine model, we considered two hypotheses: hypothesis one assumes that receptor complexes (*Figure 2—figure supplement 1f and g*) are internalized with rate proportional to the concentration of the species in which they are contained (e.g. different dimer types), and hypothesis 2, that receptor complexes are internalized with rate proportional to the product of the concentration of the species in which they are contained and the sum of the concentrations of free cytoplasmic phosphorylated STAT1 and STAT3. Hypothesis 2 is consistent with a negative feedback mechanism in which pSTAT molecules translocate to the nucleus, where they increase the translation of negative feedback proteins such as SOCS3. In order not to increase the complexity of the mathematical models with additional variables and parameters, we chose to include the feedback mechanism in hypothesis two implicitly, rather than by considering new equations for species such as SOCS3. Details of the two model hypotheses are described in the Materials and methods (Mathematical models).

We first carried out a structural identifiability analysis (*Castro and de Boer, 2020*) for the IL-27 and HypIL-6 mathematical models under both hypotheses, to determine which of the model parameters could be independently inferred. We found that all model parameters and initial concentrations were structurally identifiable, given the data and the known experimental initial concentrations. We made use of the RPE1 experimental data set to carry out Bayesian model selection for the two different hypotheses. We found that hypothesis 1 could explain the data better than hypothesis 2, with a probability of 99%. This result can be seen in *Figure 2b*, in which we plot, for different values of the distance threshold, $\delta$, between the mathematical model output and the data (see Mathematical models and Bayesian inference in Materials and methods, for details), the relative probability of each hypothesis, where hypothesis 1 is denoted $H_1$ and hypothesis 2 is denoted $H_2$. It can be observed that for smaller values of the distance threshold, which indicate data and mathematical model results are closer, the relative probability of hypothesis 1 is higher than that of hypothesis 2.

We then made use of this result to explore the mathematical models for both cytokines under hypothesis 1, in particular we performed parameter calibration. To this end (and as described in Materials and methods under Mathematical models and Bayesian inference), we carried out Bayesian inference (ABC-SMC *Toni et al., 2009*) together with the mathematical models (hypothesis 1) and the experimental data sets to quantify the reaction rates (*Figure 2—figure supplement 1*) and initial molecular concentrations (see Table 2 and Table 3). The Bayesian parameter calibration of the two models of cytokine signaling allows one to quantify the observed kinetics of pSTAT1/3 phosphorylation induced by HypIL-6 and IL-27 in RPE1 and Th-1 cells (*Figure 2c*). Substantial differences in STAT association rates to and dissociation rates from the dimeric complexes were inferred to critically contribute to defining pSTAT1/3 kinetics. *Figure 2d* shows the kernel density estimates (KDEs) for the posterior distributions of the rate constants and initial concentrations in the models. $k_{ia}^+$ denotes the rate at which STATi binds to GP130 and $k_{ib}^+$ denotes the rate at which STATi binds to IL-27R$\alpha$, for $i \in \{1,3\}$. Our results indicate that STAT1 and STAT3 exhibit different binding preferences toward IL-27R$\alpha$ and GP130. While STAT1 exhibits stronger binding to IL-27R$\alpha$ than GP130 ($k_{1b}^+ > k_{1a}^+$), STAT3 exhibits stronger binding to GP130 than IL-27R$\alpha$, ($k_{3a}^+ > k_{3b}^+$) in agreement with previous observations (*17*). To further quantify these differences in the posterior distributions for the STAT binding rates, *Table 1* provides summary statistics for each of these rates. For example, from the table we see that there is a difference of at least one order of magnitude between the means and medians of the pairs of STAT/receptor binding parameters ($k_{ia}^+$ compared with $k_{ib}^+$ for $i \in \{1,3\}$). Finally, we note that, although there exist some significant correlations between posterior distributions for pairs of

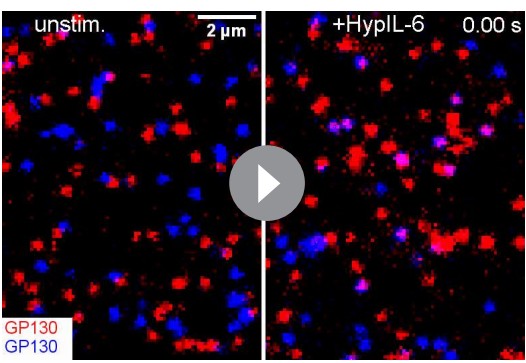

**Video 4.** Ligand-induced homodimerization of GP130. Overlay of the two spectral channels (mXFPm-GP130^RHO11: Red, mXFPm-GP130^DY649: Blue) in absence (left) or presence (right) of HypIL-6 (20 nM). Yellow lines indicate co-locomotion of IL-27Rα and GP130 (≥10 steps). Acquisition frame rate: 30 Hz, Playback: real time.

https://elifesciences.org/articles/66014#video4

parameters in the analysis, for all four pairs of STAT/receptor interaction parameters (second row of *Figure 2d*), the absolute value of the correlation coefficient is less than 0.25, and thus, the posterior distributions are representative of the rates they correspond to.

## IL-27Rα cytoplasmic domain is required for sustained pSTAT1 kinetics

The Bayesian inference carried out with the experimental data and the mathematical models clearly indicated statistically significant differences in the binding rates of STAT1/STAT3 to GP130 and IL-27Rα, to account for the different phosphorylation kinetics exhibited by HypIL-6 and IL-27. Thus, we next investigated whether the more sustained STAT1 activation by IL-27 resulted from its specific engagement of IL-27Rα. For that, we used RPE1 cells, which do not express IL-27Rα (*Figure 1—figure supplement 2a*), to systematically dissect the contribution of the IL-27Rα cytoplasmic domain to the differential pSTAT activation by IL-27. IL-27Rα's intracellular domain is very short and only encodes two Tyr susceptible to be phosphorylated in response to IL-27 stimulation, that is, Tyr543 and Ty613 (*Figure 3a*). We mutated these two Tyr to Phe to analyse their contribution to IL-27 induced signaling. We stably expressed WT IL-27Rα as well as different IL-27Rα Tyr mutants in RPE1 cells with comparable cell surface expression levels (*Figure 3b*). As the endogenous GP130 expression levels remain unaltered, all generated clones exhibited very comparable responses to HypIL-6 (*Figure 3b*, bottom panels). IL-27 triggered comparable levels of STAT1 and STAT3 activation in RPE1 cells reconstituted with IL-27Rα WT and IL-27Rα Y543F mutant, suggesting that this Tyr residue does not contribute to signaling by this cytokine (*Figure 3b* and *Figure 3—figure supplement 1a*). In RPE1 cells reconstituted with the IL-27Rα Y613F or Y543F-Y613F mutants, IL-27 stimulation resulted in 80% of the STAT3 activation, but only 20% of the STAT1 activation levels induced by this cytokine relative to IL-27Rα WT (*Figure 3b*; *Pradhan et al., 2010*). These observations suggest a tight coupling of STAT phosphorylation to one of the receptor chains; namely, IL-27Rα with pSTAT1 and GP130 with pSTAT3, respectively. We next tested how the cytoplasmic domains of GP130 and IL-27Rα shape the pSTAT kinetic profiles. Thus, we generated a stable RPE1 clone expressing a chimeric construct comprised of the extracellular and transmembrane domain of IL-27Rα but the cytoplasmic domain of GP130 (*Figure 3c*, *Figure 3—figure supplement 1b*). Again, as both cell lines express unaltered endogenous GP130 levels, they exhibited comparable responses to HyIL-6 (*Figure 3c*). Strikingly, this domain-swap resulted in a transient pSTAT1 kinetic response by IL-27 comparable to HypIL-6 stimulation. STAT3 activation on the other hand remained unaltered suggesting that the cytoplasmic domain of IL-27Rα is essential for a sustained pSTAT1 response but not for pSTAT3.

Two plausible scenarios could explain the observed pSTAT1/3 activation differential by HypIL-6 and IL-27: (i) IL-27Rα-JAK2 complex phosphorylates STAT1 faster than GP130-JAK1 complex or (ii) pSTAT1 is more quickly dephosphorylated in the IL-6/GP130 receptor homodimer. In the latter case, pSTAT deactivation by constitutively expressed phosphatases could be an additional factor of regulation. Indeed, SHP-2 has been described to bind to GP130 and shape IL-6 responses (*Kim et al., 1998*). However, our Bayesian inference results (together with the mathematical models and the experimental data) identified the STAT/receptor association rates as the only rates that could account for the greater and more sustained activation of STAT1 by IL-27. We note (as described in the Materials and methods) that the phosphorylation rate, denoted by q, of STAT1 and STAT3 when bound to a dimer (homo- or hetero-) has been assumed to be independent of the STAT type and the receptor chain. Moreover, the model also included dephosphorylation of free pSTAT molecules, and predicted that the rates at which these reactions occur ($d_1$ and $d_3$) had rather similar posterior

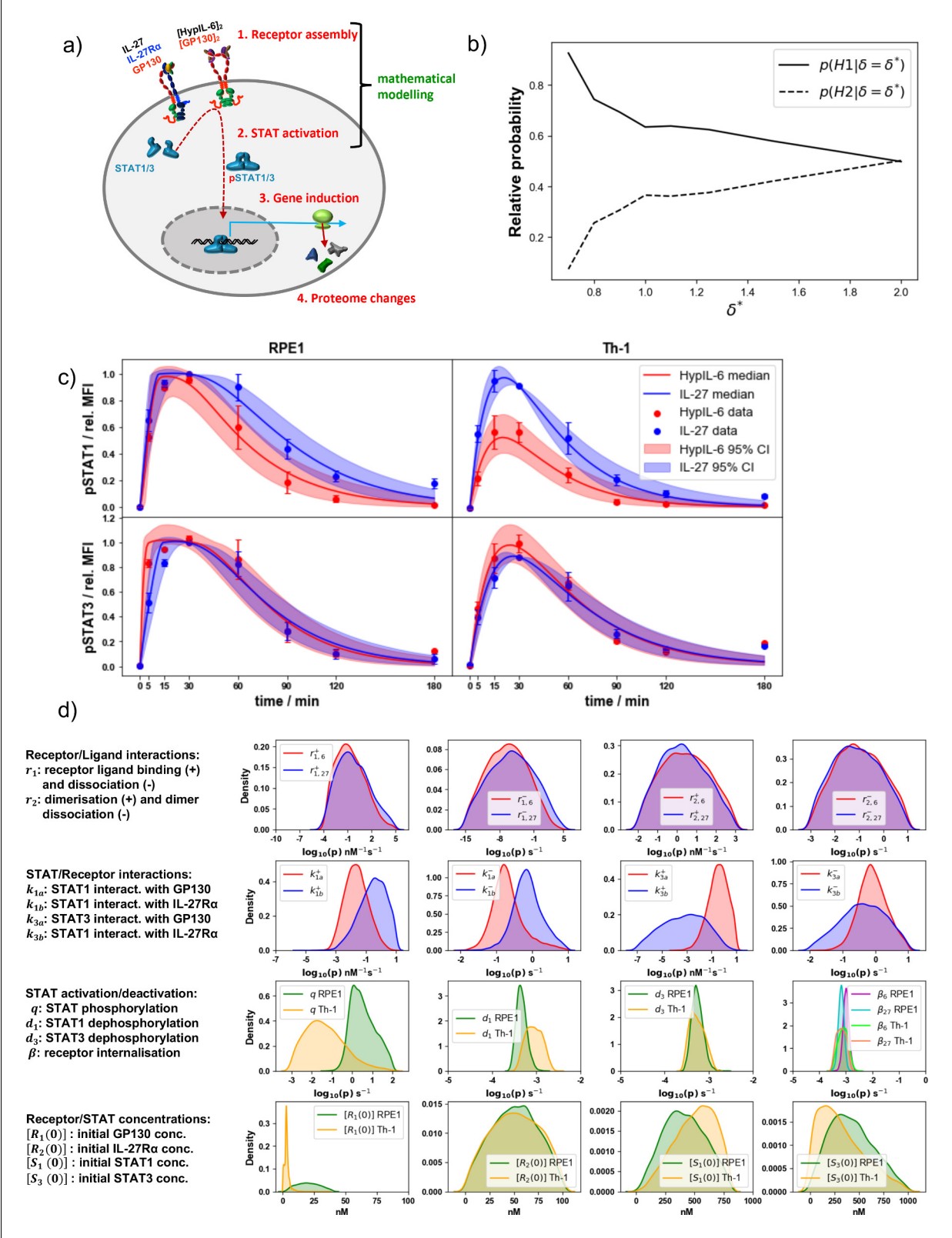

**Figure 2.** Mathematical modeling results in RPE1 and Th-1 cells. (**a**) Simplified cartoon model of IL-27/HypIL-6 signal propagation layers and coverage of the mathematical modeling approach. (**b**) Model selection results showing the relative probabilities of each hypothesis, for different values of the distance threshold, $\delta^*$, in RPE1 cells. (**c**) Pointwise median and 95% credible intervals of the predictions from the mathematical model, calibrated with the experimental data, using the posterior distributions for the parameters from the ABC-SMC. For the experimental data, phosphorylation kinetics of

*Figure 2 continued on next page*

Figure 2 continued

pSTAT1 and pSTAT3 were followed in RPE1 IL-27Rα after stimulation with saturating concentrations of IL-27 (2 nM) and HypIL-6 (10 nM). (d) Kernel density estimates of the posterior distributions for the parameters $p \in \left\{ r_{1,j}^+, r_{1,j}^-, r_{2,j}^+, r_{2,j}^-, k_{ia}^+, k_{ia}^-, k_{ib}^+, k_{ib}^-, q, d_i, \beta_j, [R_1(0)], [R_2(0)], [S_1(0)], [S_3(0)] \right\}$ in the mathematical models where $j \in \{6, 27\}$ and $i \in \{1, 3\}$. A '+' in the parameter notation indicates that this is a forward rate constant (binding) and a '−' in the notation indicates that this is a reverse rate constant (dissociation). In the first row, a '6' in the notation indicates that this is a reaction rate in the HypIL-6 system and a '27' indicates that this is a reaction rate in the IL-27 system.

The online version of this article includes the following figure supplement(s) for figure 2:

**Figure supplement 1.** Schematic model of involved reactions and parameters for IL-27 and HypIL-6 receptor activation.

distributions, hence arguing against the potential role of phosphatases to specifically target STAT1 upon HypIL-6 stimulation. To distinguish between the two plausible scenarios, we next determined the rates of pSTAT1/3 dephosphorylation by blocking JAK activity upon cytokine stimulation making use of the JAK inhibitor Tofacitinib in RPE1 cells. Tofacitinib was added 15 min after stimulation with either cytokine and pSTAT1 and pSTAT3 levels were measured at the indicated times. JAK inhibition markedly shortened the pSTAT1/3 activation profiles induced by both cytokines (*Figure 3d*, *Figure 3—figure supplement 1c*). The relative dephosphorylation rates could then be determined by the signal intensity ratio of +/- Tofacitinib. Even though pSTAT1 levels were more affected by JAK inhibition than those of pSTAT3, the observed relative changes were nearly identical for IL-27 and HypIL-6. These findings were also confirmed for Th-1 cells (*Figure 3—figure supplement 1d and e*) and indicate, that selective phosphatase activity cannot serve as an explanation for the pSTAT1/3 differential by HypIL-6 and IL-27, in agreement with our mathematical modeling predictions. Similarly, we tested whether neosynthesis of feedback inhibitors such as SOCS3 (*Croker et al., 2003*) would selectively impair signaling by HypIL-6 but not by IL-27. To this end we pre-treated cells with Cycloheximide (CHX) and followed the pSTAT1/3 kinetics induced by the two cytokines (*Figure 3—figure supplement 2a and b*). CHX treatment resulted in more sustained pSTAT3 activity for both cytokines. To our surprise, STAT1 phosphorylation by IL-27 was even more sustained while pSTAT1 levels induced by IL-6 remained unaffected. These observations exclude that feedback inhibitors selectively impair STAT1 activation kinetics by HypIL-6 and thus do not account for the faster STAT1 dephosphorylation kinetics observed under HypIL-6 stimulation. Overall, our data from the chimera and mutant experiments, which were not used in the Bayesian parameter calibration, provide additional support: they validate the mathematical models of HypIL-6 and IL-27 signaling and point to the differential association/dissociation of STAT1 and STAT3 to IL-27Rα and GP130, respectively, as the main factor defining STAT phosphorylation kinetics in response to HypIL-6 and IL-27 stimulation.

## Unique and overlapping effects of IL-27 and HypIL-6 on the Th-1 phosphoproteome

Thus far, we have investigated the differential activation of STAT1/STAT3 induced by HypIL-6 and IL-27. Next, we asked whether IL-27 and IL-6 induced the activation of additional and specific

**Table 1.** Summary statistics of the posterior distributions for the STAT/receptor binding and dissociation rates.

| Parameter | Mean | Median |
|---|---|---|
| $k_{1a}^+$ | $1.1 \times 10^{-1}$ | $2.2 \times 10^{-2}$ |
| $k_{1a}^-$ | $4.2 \times 10^{-1}$ | $1.8 \times 10^{-1}$ |
| $k_{1b}^+$ | $1.2 \times 10^0$ | $3.4 \times 10^{-1}$ |
| $k_{1b}^-$ | $1.2 \times 10^0$ | $7.2 \times 10^{-1}$ |
| $k_{3a}^+$ | $1.1 \times 10^0$ | $3.7 \times 10^{-1}$ |
| $k_{3a}^-$ | $1.4 \times 10^0$ | $8.0 \times 10^{-1}$ |
| $k_{3b}^+$ | $6.8 \times 10^{-2}$ | $7.5 \times 10^{-4}$ |
| $k_{3b}^-$ | $1.2 \times 10^0$ | $4.3 \times 10^{-1}$ |

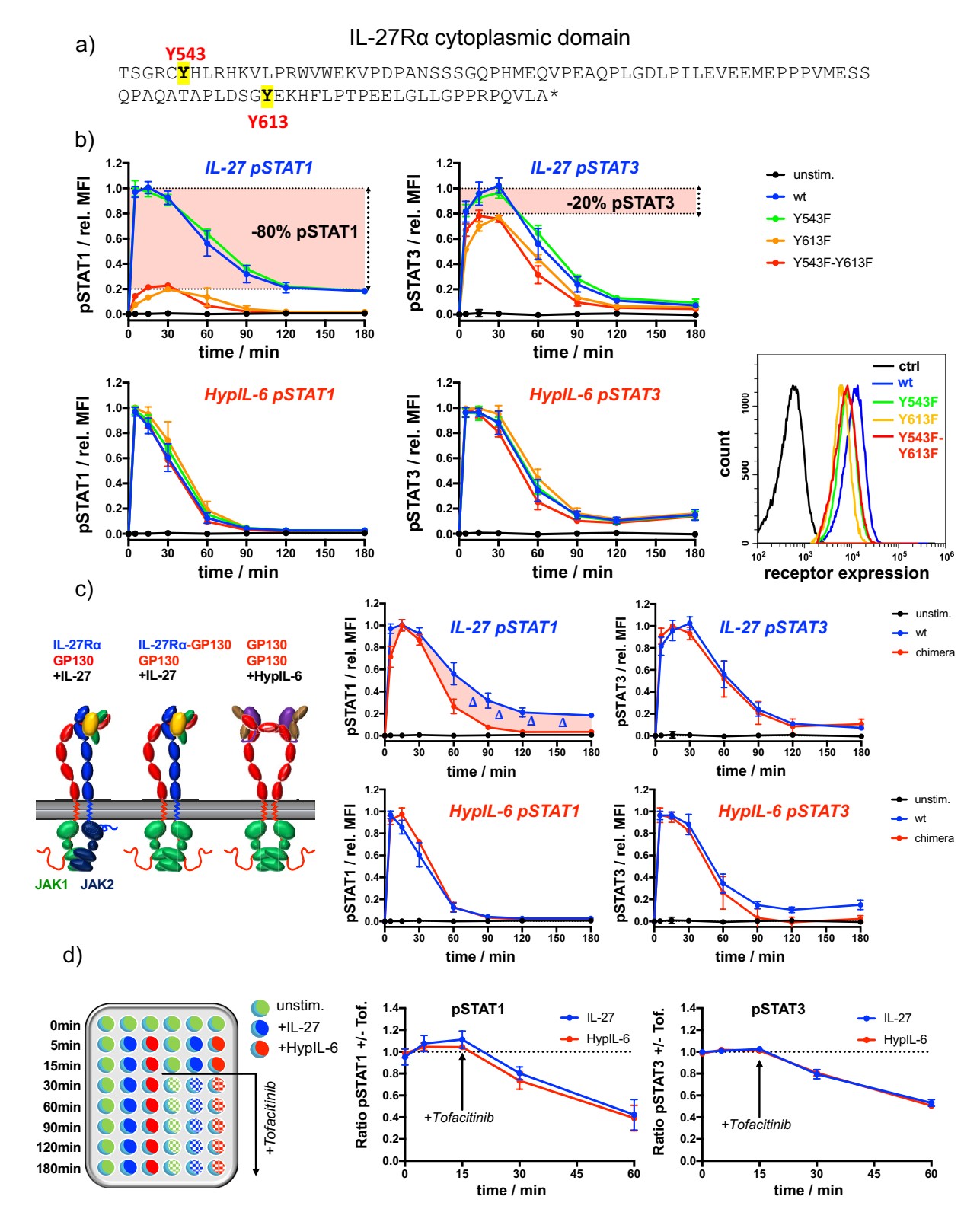

**Figure 3.** IL-27Rα cytoplasmic domain is required for sustained pSTAT1 kinetics. (a) Representation of the cytoplasmic domain of IL-27Rα with its highlighted tyrosine residues Y543 and Y613. (b) STAT1 and STAT3 phosphorylation kinetics of RPE1 clones stably expressing wt and mutant IL-27Rα after stimulation with IL-27 (10 nM, top panels) or after stimulation with HypIL-6 (20 nM, bottom panels), normalized to maximal levels of wt IL-27Rα stimulated with IL-27 (top) or HypIL-6 (bottom). Data was obtained from three experiments with each two technical replicates, showing mean ± std dev.

*Figure 3 continued on next page*

Figure 3 continued

Bottom right: cell surface levels variants measured by flow cytometry for indicated IL-27Rα cell lines. (c) Cytoplasmic domain of IL-27Rα is required for sustained pSTAT1 activation. Left: Cartoon representation of receptor complexes. Right: STAT1 and STAT3 phosphorylation kinetics of RPE1 clones stably expressing wt IL-27Rα and IL-27Rα-GP130 chimera after stimulation with IL-27 (10 nM, top panels) or after stimulation with HypIL-6 (20 nM, bottom panels). Data was normalized to maximal levels for each cytokine and cell line. Data was obtained from two experiments with each two technical replicates, showing mean ±std dev. (d) Phosphatases do not account for differential pSTAT1/3 activity induced by IL-27 and HypIL-6. Left: Schematic representation of workflow using JAK inhibitor Tofacitinib. Right: MFI ratio of Tofacitinib-treated and non-treated RPE1 mXFPe-IL-27Rα cells for pSTAT1 and pSTAT3 after stimulation with IL-27 (10 nM) and HypIL-6 (20 nM). Data was obtained from two experiments with each two technical replicates, showing mean ± std dev.

The online version of this article includes the following figure supplement(s) for figure 3:

**Figure supplement 1.** Characterization of IL-27Rα mutants and probing phosphatase activity by Tofacitinib-mediated inhibition of JAK/STAT signaling.

**Figure supplement 2.** Effects of Cycloheximide-mediated inhibition of protein neosynthesis on IL-27 and HypIL-6 signaling.

intracellular signaling programs that could contribute to their unique biological profiles. To this end, we investigated the IL-27 and HypIL-6 activated signalosome using quantitative mass-spectrometry-based phospho-proteomics. MACS-isolated CD4+ were polarized into Th-1 cells and expanded in vitro for stable isotope labeling by amino acids in cell culture (SILAC). Cells were then stimulated for 15 min with saturating concentrations of IL-27 (10 nM), HypIL-6 (20 nM) or left untreated. Samples were enriched for phosphopeptides (Ti-IMAC), subjected to mass spectrometry and raw files analysed by MaxQuant software (*Figure 4—figure supplement 1*). In total we could quantify ~6400 phosphopeptides from 2600 proteins, identified across all conditions (unstimulated, IL-27, HypIL-6) for at least two out of three tested donors. For IL-27 and HypIL-6 we detected similar numbers of significantly upregulated (87 vs. 78) and downregulated (155 vs. 140) phosphorylation events (*Figure 4a*) and systematically categorized them in context with their cellular location and ascribed biological functions (*Figure 4—figure supplement 1b and c*; *Huang et al., 2009b*). The two cytokines shared approximately half of the upregulated and one third of the downregulated phospho-peptides (*Figure 4—figure supplement 2a*) but also exhibited differential target phosphorylation (*Figure 4b* and *Figure 4—figure supplement 2b*). As expected, we found multiple members of the STAT protein family among the top phosphorylation hits by the two cytokines, validating our study (*Figure 4b and c*). In line with our previous observations, we detected the same relative amplitudes for tyrosine phosphorylated STAT3 and STAT1. In addition to tyrosine-phosphorylation, we detected robust serine-phosphorylation on S727 for STAT1 and STAT3 (*Figure 4c*). While pS-STAT1 activity correlated with pY-STAT1 with IL-27 being more potent than HypIL-6, this was not the case for STAT3. Despite an identical pY-STAT3 phosphorylation profile, HypIL-6 induced a ~50% higher pS-STAT3 relative to IL-27 (*Figure 4c*). These results were corroborated, following the phosphorylation kinetics of pS-STAT1 and pS-STAT3 by flow-cytometry (*Figure 4d*). Interestingly, while IL-27 and IL-6 have been described to also activate other signaling pathways, such as MAPK, p38, and AKT (*Diegelmann et al., 2012*), we did not observe this in our phospho-target screening.

Given the overlapping phospho-proteomic changes, gene ontology (GO) analysis associated several sets of phosphopeptides with biological processes that were mostly shared between both cytokines (*Figure 4e*, *Figure 4—figure supplement 2c*). A large set of phospho-peptides was linked to transcription initiation (including JAK/STAT signaling) or mRNA modification (*Figure 4e*). Interestingly, IL-27 stimulation was associated to negative regulation of RNA polymerase II, whereas a positive regulation was detected for HypIL-6. A closer look into the functional regulation of RNA-pol II activity by the two cytokines revealed that multiple proteins involved in this process were differentially regulated by HypIL-6 and IL-27 (*Figure 4f*). While positive regulators of RNA-pol II transcription, such as Negative Elongation Factor A (NELFA), PPM1G, RCHY1 and POL2RA, were much more phosphorylated in response to HypIL-6 than IL-27, negative regulators of RNA-pol II transcription, such as LARP7, were much more engaged by IL-27 treatment than by HypIL-6 (*Figure 4f*). Interestingly, in a previous study we linked RNA-pol II regulation with the levels of STAT3 S727phosphorylation induced by HypIL-6 via recruitment of CDK8 to STAT3-dependent genes (*Bancerek et al., 2013*). Our phospho-proteomic analysis thus, suggests that IL-27 and HypIL-6 recruit different transcriptional complexes that ultimately could contribute to provide gene expression specificity by the two cytokines. Additionally, we identified several interesting IL-27-specific phosphorylation targets. One example was Ubiquitin Protein Ligase E3 Component N-Recognin 5 (UBR5). Phosphorylated

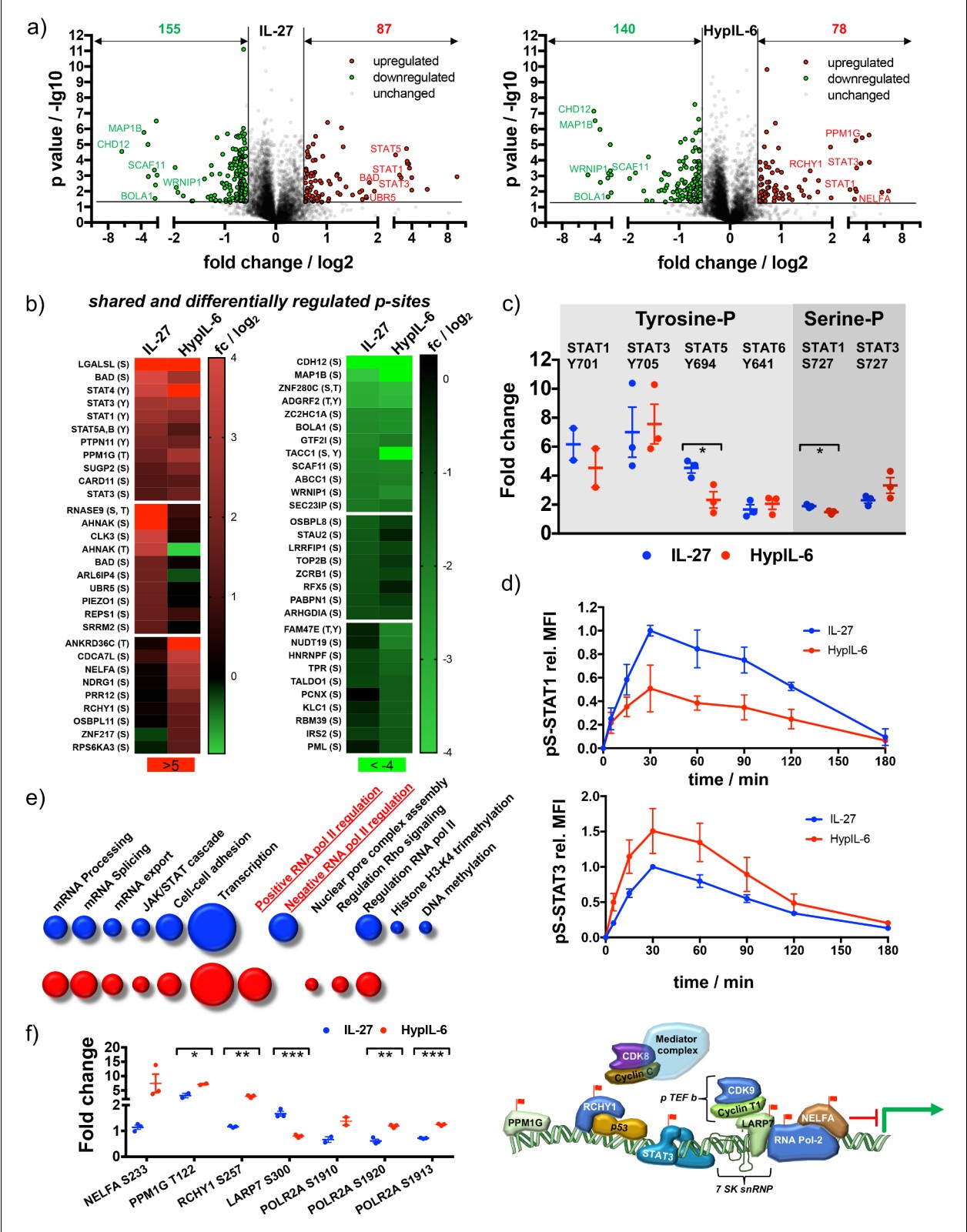

**Figure 4.** Unique and overlapping effects of IL-27 and HypIL-6 on the phosphoproteome of Th-1 cells. (**a**) Volcano plot of the phospho-sites regulated (p value ≤ 0.05, fold change ≥+1.5 or≤−1.5) by IL-27 (left) and HypIL-6 (right). Cells were stimulated for 15 min at saturating concentrations (IL-27: 10 nM, HypIL-6: 20 nM). Data was obtained from three biological replicates. (**b**) Heatmap representation (examples) of shared and differentially up- (left) and downregulated (right) phospho-sites after IL-27 and HypIL-6 stimulation. Data represents the mean (log₂) fold change of three biological replicates.

*Figure 4 continued on next page*

*Figure 4 continued*

(c) Tyrosine and Serine phosphorylation of selected STAT proteins after stimulation with IL-27 (red) and HypIL-6 (blue). Two-tailed Student's T-test: *p<0.05, **p≤0.01,***p≤0.001; n.s., not significant. (d) pS727-STAT1 and pS727-STAT3 phosphorylation kinetics in Th-1 cells after stimulation with IL-27 or HypIL-6, normalized to maximal IL-27 stimulation. Data was obtained from three biological replicates with each two technical replicates, showing mean ±std dev. (e) GO analysis 'biological processes' of the phospho-sites regulated by IL-27 (red) and HypIL-6 (blue) represented as bubble-plots. (f) Phosphorylation of target proteins associated with STAT3/CDK transcription initiation complex after stimulation with IL-27 (blue) and HypIL-6 (red) and schematic representation of transcription regulation of RNA polymerase II with identified phospho-sites (red flags). Two-tailed Student's T-test: *p<0.05, **p≤0.01,***p≤0.001; n.s., not significant.

The online version of this article includes the following figure supplement(s) for figure 4:

**Figure supplement 1.** Workflow for phospho-proteomics and cellular localization of identified phosphorylation targets.

**Figure supplement 2.** Phospho-proteomics - shared and unique targets and gene ontology analysis.

UBR5 leads to ubiquitination and subsequent degradation of Rorγc (*Rutz et al., 2015*), the key transcription factor required for Th-17 lineage commitment, thus limiting Th-17 differentiation (*Figure 4—figure supplement 2d*). A second example is PAK2, which phosphorylates and stabilizes FoxP3 leading to higher levels of T_{Reg} cells (*Figure 4—figure supplement 2d*; *O'Hagan et al., 2017*). Moreover, IL-27 stimulation led to a very strong phosphorylation of BCL2-associated agonist of cell death (BAD), a critical regulator of T-cell survival and a well-known substrate of the PAK2 kinase (*Ye and Field, 2012*). Overall, our data show a large overlap between the IL-6 and IL-27 signaling program, with a strong focus on JAK/STAT signaling. However, IL-27 engages additional signaling intermediaries that could contribute to its unique immuno-modulatory activities. Further studies will be required to assess how these IL-27 specific signaling pockets contribute to shape IL-27 responses.

## Kinetic decoupling of gene induction programs depends on sustained STAT1 activation and IRF1 expression by IL-27

Next, we investigated how the different kinetics of STAT activation induced by HypIL-6 and IL-27 ultimately modulated gene expression by these two cytokines. To this end, we performed RNA-seq analysis of Th-1 cells stimulated with HypIL-6 or IL-27 for 1 hr, 6 hr and 24 hr to obtain a dynamic perspective of gene regulation. We identified ~12500 shared genes that could be quantified for all three donors and throughout all tested experimental conditions. In a first step, we compared how similar the gene programs induced by HypIL-6 and IL-27 were. Principal component analysis (PCA) was run for a subset of genes, found to be significantly up- (total ~250) or downregulated (total ~950) by either of the experimental conditions (p value≤0.05, fold change ≥+2 or ≤−2). At one hour of stimulation HypIL-6 and IL-27 induced very similar gene programs, with the two cytokines clustering together in the PCA analysis regardless of whether we focused on the subsets of upregulated or downregulated genes (*Figure 5a*). However, the similarities between the two cytokines changed dramatically in the course of continuous stimulation. While the two cytokines induced the downregulation of comparable gene programs at 6 hr and 24 hr stimulation, as denoted by the close clustering in the PCA analysis (*Figure 5a*, right panel) and the fraction of shared genes (~40%, *Figure 5b*, *Figure 5—figure supplement 1a–c*, *Figure 5—figure supplement 2a*), this was not observed for upregulated genes. Although the two cytokines induced comparable gene upregulation programs after 1 hr of stimulation (~80% shared genes), this trend almost completely disappeared at later stimulation times (*Figure 5a and b*, *Figure 5—figure supplement 2b*). This is well-reflected by the absolute numbers of up- or downregulated genes observed for IL-27 and HypIL-6 (*Figure 5c*). Stimulation with both cytokines yielded a similar trend of gene downregulation (*Figure 5c*, right panel). However, while HypIL-6 stimulation resulted in a spike of gene upregulation at 1 hr that quickly disappeared at later stimulation times, IL-27 stimulation was capable to increase the number of upregulated genes beyond 6 hr of stimulation and maintains it even after 24 hr (*Figure 5c*, left panel). This 'kinetic decoupling' of gene induction seems to have a striking functional relevance. Gene set enrichment analysis (GSEA) (*Liao et al., 2019*) identified several reactome pathways to be enriched for IL-27 over the course of stimulation – most of them linked with Interferon signaling and immune responses (*Figure 5d*). In contrast, for HypIL-6 stimulation no pathway enrichment was detected. Most importantly, the vast majority of IL-27-induced genes that were associated to these pathways belonged to genes upregulated by IL-27 treatment and that have been previously

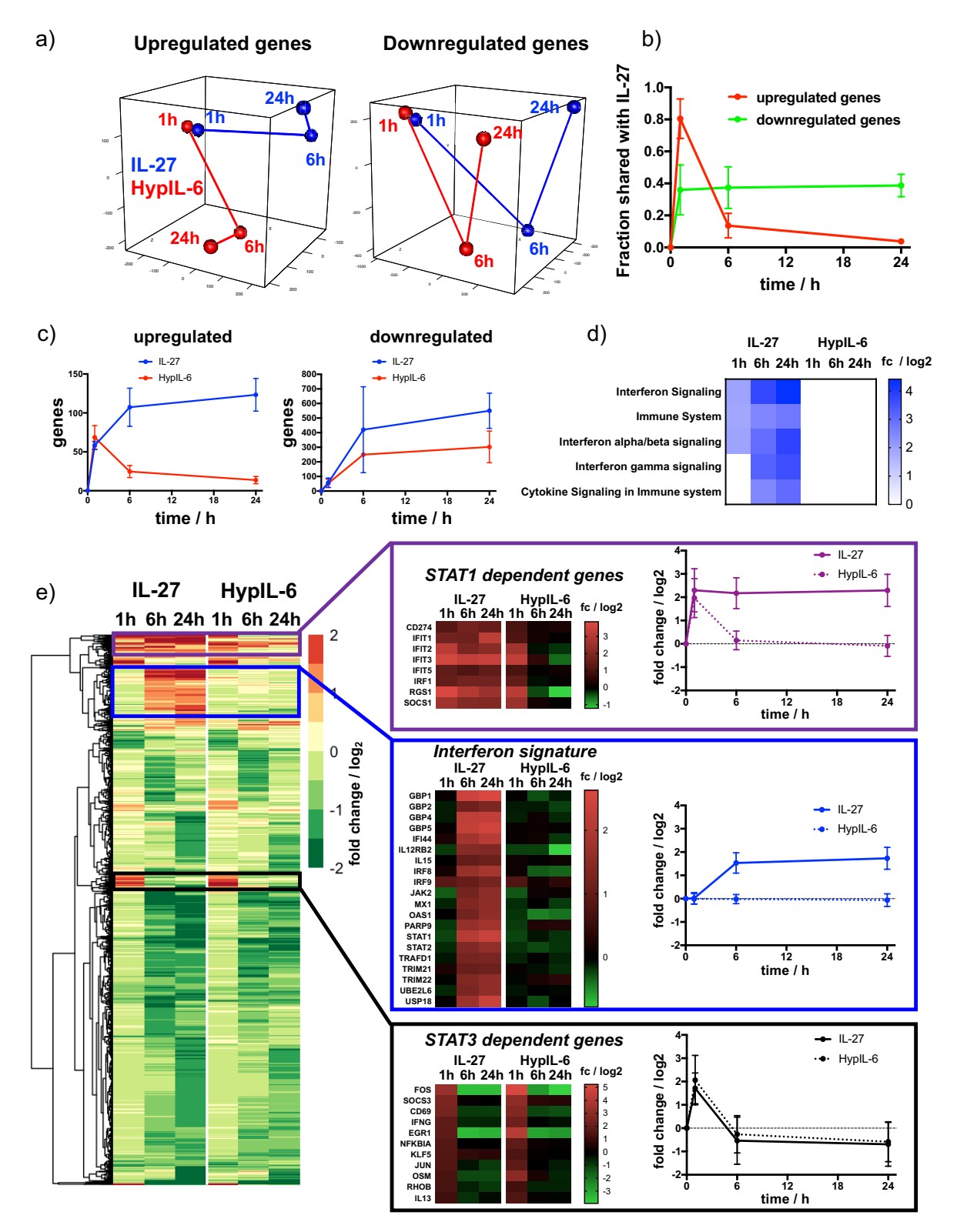

**Figure 5.** Kinetic decoupling of gene induction programs depends on sustained STAT1 activation by IL-27. (a) Principal component analysis for genes found to be significantly upregulated (left) or downregulated (right) for at least one of the tested conditions (time and cytokine). Data was obtained from three biological replicates. (b) Kinetics of gene induction shared between IL-27 and HypIL-6 (relative to IL-27) for upregulated genes (red) or downregulated genes (green). (c) Kinetics of gene numbers induced after IL-27 and HypIL-6 stimulation for upregulated genes (left) and downregulated

*Figure 5 continued on next page*

*Figure 5 continued*

genes (right). (**d**) GSEA reactome analysis of selected pathways with significantly altered gene induction in response to IL-27 or HypIL-6 stimulation. Data represents the mean (log$_2$) fold change of three biological replicates. (**e**) Cluster analysis comparing the gene induction kinetics after IL-27 or HypIL-6 stimulation. Gene induction heatmaps for example genes as well as induction kinetics (mean) are shown for highlighted gene clusters. Data represents the mean (log$_2$) fold change of three biological replicates.

The online version of this article includes the following figure supplement(s) for figure 5:

**Figure supplement 1.** IL-27 and HypIL-6 stimulated gene induction profiles represented as volcano plots.

**Figure supplement 2.** Kinetic decoupling of gene induction programs - shared and unique genes and GSEA pathway analysis.

**Figure supplement 3.** IL-27-induced upregulation of IRF1 amplifies induction of STAT1-dependent genes.

linked to STAT1 activation (*Satoh and Tabunoki, 2013*; *Rusinova et al., 2013*; *Figure 5—figure supplement 2c*). Although HypIL-6 treatment resulted in the induction of some of these genes, their expression was very transient in time, in agreement with the short STAT1 activation kinetic profile exhibited by HypIL-6 (*Figure 5—figure supplement 2b and c*).

Next, we performed cluster analysis to find further similarities and discrepancies between the gene expression programs engaged by HypIL-6 and IL-27 (*Figure 5e*). Since genes downregulated by IL-27 and HypIL-6 showed overall good similarity throughout the whole kinetic series, we mainly focused on differences in upregulated gene induction. We identified three functionally relevant gene clusters. The first gene cluster corresponds to genes that are transiently and equally induced by HypIL-6 and IL-27. These genes peak after one hour and return to basal levels after 6 hr and 24 hr of stimulation (*Figure 5e*). Interestingly, this cluster contains classical IL-6-induced and STAT3-dependent genes, such as members of the NFκB and Jun/Fos transcriptional complex (*Suh et al., 2008*), as well as the feedback inhibitor Suppressor Of Cytokine Signaling 3 (*SOCS3*) (*Villarino et al., 2006*) and T-cell early activation marker CD69. (*Figure 5e*). A second cluster of genes corresponded to genes that were persistently activated by IL-27 but only transiently by HypIL-6 (*Figure 5e*). Among these genes we found classical STAT1-dependent genes, such as *SOCS1*, Programmed Cell Death Ligand 1 (*PDL1 = CD274*) (*Hirahara et al., 2012*) and members of the interferon-induced protein with tetratricopeptide repeats (IFIT) family. The third cluster of genes corresponded to genes exhibiting strong and sustained activation by IL-27 after 6 hr and 24 hr stimulation but no activation by HypIL-6 at all. This '2nd wave' of gene induction by IL-27 was almost exclusively comprised of classical Interferon Stimulated Genes (ISGs) (*Figure 5—figure supplement 2c*), such as *STAT1 and 2*, Guanylate Binding Protein 1 (*GBP1*), *GBP2, 4 and 5*, and *IRF8 and 9*.

It is worth mentioning, that genes in the third cluster appear to require persistent STAT1 activation (*Hu et al., 2002*; *Francois-Newton et al., 2012*) and were the basis for the IFN signature identified in our reactome pathway analysis. Still, we were surprised about the magnitude of this 2nd gene wave. Even though IL-27 exerts a sustained pSTAT1 kinetic profile, pSTAT1 levels were down to ~10% of maximal amplitude after 3 hr of stimulation. We reasoned that additional factors could further amplify the STAT1 response for IL-27 but not for HypIL-6. Within the 1st wave of STAT1-dependent genes, we also spotted the transcription factor Interferon Response Factor 1 (*IRF1*), that was continuously induced throughout the kinetic series in response to IL-27 but only transiently spiking after 1 hr of HypIL-6 stimulation (*Figure 5e*). IRF1 expression was shown to prolong pSTAT1 kinetics (*Zenke et al., 2018*) and to be required for IL-27-dependent Tr-1 differentiation and function (*Karwacz et al., 2017*). We confirmed the kinetics of IRF1 protein expression by flow cytometry and showed higher and more sustained protein levels after IL-27 stimulation relative to HypIL-6 (*Figure 5—figure supplement 3a*). Next, we tested in our RPE1 IL-27Rα cell system, whether siRNA-mediated knockdown of IRF1 would alter the gene induction profiles of certain STAT1 or STAT3-dependent marker genes. In RPE1 IL-27Rα cells, IRF1 protein levels were peaking around 6 hr after stimulation with IL-27 and transfection with IRF1-targeting siRNA knocked down expression by >80% (*Figure 5—figure supplement 3b*). Importantly, knockdown of IRF1 did not alter the overall kinetics of pSTAT1 and pSTAT3 activation (*Figure 5—figure supplement 3c*). Induction of STAT1-dependent genes *STAT1*, *GBP5*, and *OAS1* as well as STAT3-dependent gene *SOCS3* were followed by RT qPCR (*Figure 5—figure supplement 3d*). Interestingly, up to 6 hr of stimulation, the gene induction curves were identical for control- and IRF1-siRNA treated cells. Later than 6 hr – that is, when IRF1 protein levels are peaking – the gene induction was decreased between 40 and 70% in absence of IRF1. Strikingly, expression of *SOCS3*, a classical STAT3-dependent reporter gene was

transient and independent on IRF1 levels, highlighting that IRF1 selectively amplifies STAT1-dependent gene induction. Taken together our data support a scenario whereby IL-27 by exhibiting a kinetic decoupling of STAT1 and STAT3 activation is capable of triggering independent gene expression waves, which ultimately contribute to shape its distinct biology.

## IL-27-induced STAT1 response drives global proteomic changes in Th-1 cells

Next, we aimed to uncover how the distinct gene expression programs engaged by HypIL-6 and IL-27 ultimately relate to alterations of the Th-1 cell proteome. For that, we continuously stimulated SILAC -labeled Th-1 cells for 24 hr with saturating doses of IL-27 and HypIL-6 and compared quantitative proteomic changes to unstimulated controls (*Figure 6a*). We quantified ~3600 proteins present in all three biological replicates and in all tested conditions (unstimulated/IL-27/HypIL-6). Both cytokines downregulated a similar number of proteins (IL-27: 57, HypIL-6: 52) (*Figure 6b*) with approximately half of them being shared by the two cytokines, mimicking our observations in the RNA-seq studies (*Figure 6c*, *Figure 6—figure supplement 1*). With 68 upregulated proteins, IL-27 was almost twice as potent as HypIL-6 (35 proteins) with very little overlap.

Among the upregulated proteins by IL-27 but not HypIL-6, we detected several proteins with described immune-modulatory functions on T-cells. One of these proteins was Transforming Growth Factor β (TGF-β), which is a key regulator with pleiotropic functions on T-cells (*Yoshimura et al., 2010*). TGF-β has been identified to synergistically act with IL-27 to induce IL-10 secretion from Tr-1 cells – thus accounting for one of the key anti-inflammatory functions of IL-27 (*Awasthi et al., 2007*). On the other hand, we also found *SELPLG*-encoded protein RSGL-1 which is critically required for efficient migration and adhesion of Th-1 cells to inflamed intestines (*Brown et al., 2012*; *Matsumoto et al., 2007*). Interestingly, we found LARP7 moderately upregulated by IL-27. This negative regulator for RNA pol II was also identified in our phospho-target screening and selectively engaged by IL-27 (*Figure 4f*). IL-27 and HypIL-6 share ~60% of downregulated proteins, but without strong functional patterns. Both cytokines downregulated several proteins related to mitotic cell cycle (LIG1, CSNK2B, PSMB1) mRNA processing and splicing (NCBP2, PCBP2, NUDT21) (*Slenter et al., 2018*).

Strikingly, a significant number (~40%) of proteins upregulated by IL-27 belong to the group of ISGs (*Figure 6b and c*, *Figure 6—figure supplement 1b*). This particular set of proteins including STAT1, STAT2, MX Dynamin like GTPase 1 (MX1), Interferon Stimulated Gene 20 (ISG20) or Poly (ADP-Ribose) Polymerase Family Member 9 (PARP9) was not markedly altered by HypIL-6. Of note: the overall expression patterns of the most significantly altered proteins are congruent to the gene induction patterns observed after 6 hr and 24 hr (*Figure 6d and e*, Supp. Fig. 10b). Similar to this, GSEA reactome analysis identified again pathways associated with interferon signaling and cytokine/immune system but failed to detect any significant functional enrichment by HypIL-6 (*Figure 6e*, *Figure 6—figure supplement 1b & c*). Finally, we correlated RNAseq-based gene induction patterns with detected proteomic changes. To our surprise, we only found a relatively low number of shared hits. However, the identified proteins belong exclusively to a group upregulated by IL-27 (*Figure 6f*). They are all located in the '2nd gene wave' cluster and all of them are regulated by ISGs (*Figure 5e*). Taken together these results provide compelling evidence that sustained pSTAT1 activation by IL-27 accounts for its gene induction and proteomic profiles, thus, giving a mechanistic explanation for the diverse biological outcomes of IL-27 and IL-6. Our observations are in good agreement with previous findings in cancer cells, showing that particularly the involvement of STAT1 activation is responsible for proteomic remodeling by IL-27 (*Petretto et al., 2016*).

## Receptor and STAT concentrations determine the nature of the IL-6/IL-27 response

Our data suggest that STAT molecules compete for binding to a limited number of phospho-Tyr motifs in the intracellular domains of cytokine receptors. A direct consequence derived from this hypothesis is that cells can adjust and change their responses to cytokines by altering their concentrations of specific STATs or receptors molecules. To assess to what degree immune cells differ in their expression of cytokine receptors and STATs, we investigated levels of IL-6Rα, GP130, IL-27Rα, STAT1, and STAT3 protein expression across different immune cell populations making use of the

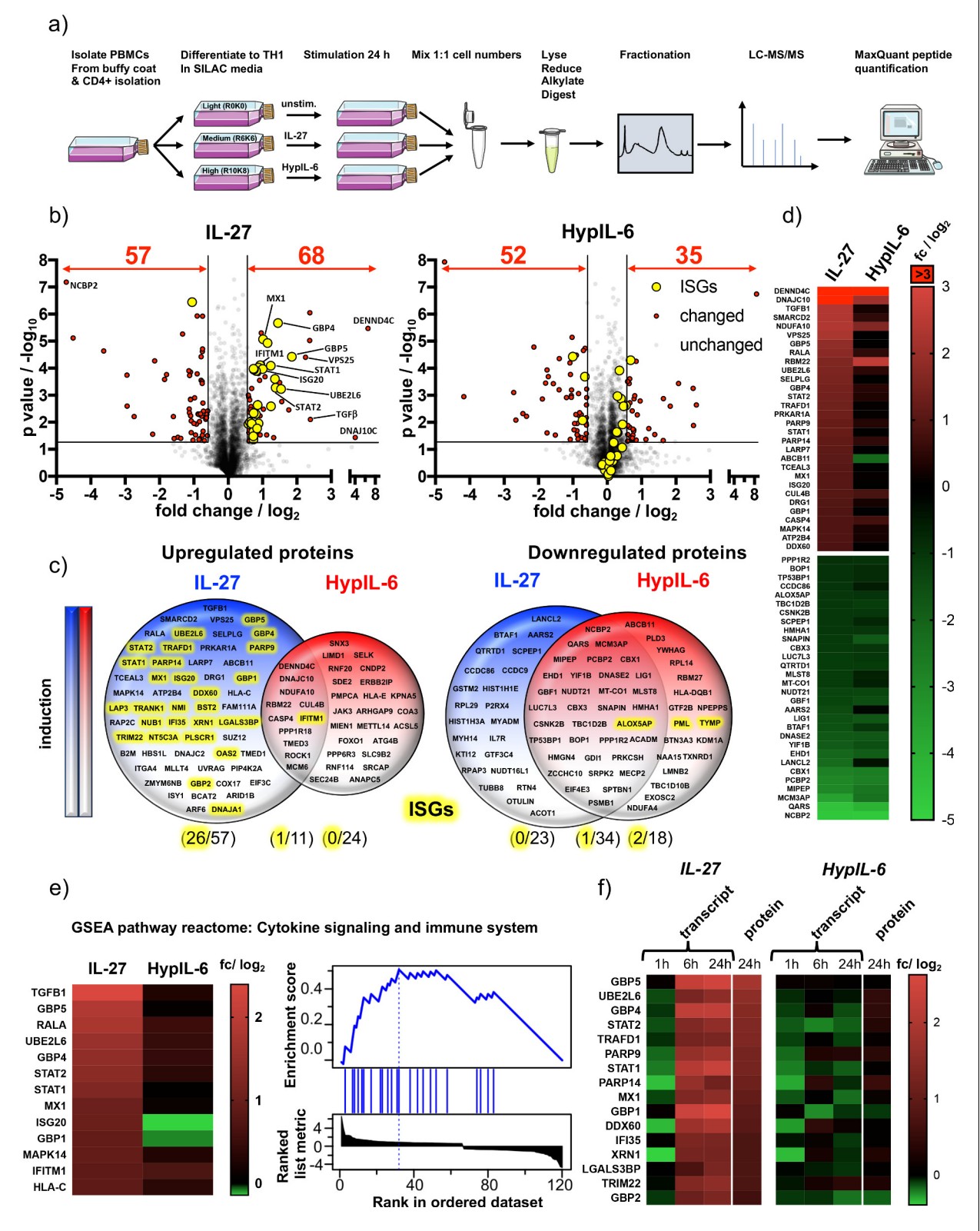

**Figure 6.** IL-27-induced STAT1 response drives global proteomic changes in Th-1 cells. (**a**) Workflow for quantitative SILAC proteomic analysis of Th-1 cells continuously stimulated (24 hr) with IL-27 (10 nM), HypIL-6 (20 nM) or left untreated. (**b**) Global proteomic changes in Th-1 cells induced by IL-27 (left) or HypIL-6 (right) represented as volcano plots. Proteins significantly up- or downregulated are highlighted in red (p value ≤ 0.05, fold change ≥+1.5 or≤−1.5). Significantly altered ISG-encoded proteins by IL-27 are highlighted in yellow. Data was obtained from three biological

*Figure 6 continued on next page*

*Figure 6 continued*

replicates. (**c**) Venn diagrams comparing unique upregulated (left) and downregulated (right) proteins by IL-27 (blue) and HypIL-6 (red) as well as shared altered proteins. ISG-encoded proteins are highlighted in yellow. (**d**) Heatmaps of the top 30 up- and downregulated proteins by IL-27 compared to HypIL-6. Data representation of the mean (log$_2$) fold change of three biological replicates. (**e**) Heatmap representation and enrichment plot of proteins identified by GSEA reactome pathway enrichment analysis 'Cytokine signaling and immune system' induced by IL-27. Data representation of the mean (log$_2$) fold change of three biological replicates. (**f**) Correlation of IL-27 and HypIL-6-induced RNA-seq transcript levels ($\geq +2$ or $\leq -2$ fold change) with quantitative proteomic data ($\geq +1.5$ or $\leq -1.5$ fold change). Data representation of the mean (log$_2$) fold change of three biological replicates.

The online version of this article includes the following figure supplement(s) for figure 6:

**Figure supplement 1.** Shared and unique proteomic changes upon IL-27 and HypIL-6 stimulation and GSEA pathway reactome analysis.

Immunological Proteomic Resource (ImmPRes - http://immpres.co.uk) database. Strikingly, the level of expression of these proteins change dramatically across the populations studied (***Figure 7a***), suggesting that these cells could potentially produce very different responses to HypIL-6 and IL-27 stimulation.

In order to quantify (and predict) how changes in expression levels of different proteins modify the kinetics of pSTAT, we made use of the two mathematical models of HypIL-6 and IL-27 stimulation and the parameters inferred with Bayesian methods. Our mathematical models could accurately reproduce the experimental results generated across our study, that is, signaling by the IL-27Rα chimeric and IL-27Rα-Y616F mutant receptors and dose/response studies (***Figure 7—figure supplement 1a–c***), making use of the posterior parameter distributions generated from the Bayesian parameter calibration. Having developed mathematical models which are able to accurately explain the experimental data (***Figure 3—figure supplement 1c and d***) and reproduce independent experiments (***Figure 3b and c***), we then sought to use the models to predict pSTAT signaling kinetics under different concentration regimes of receptors and STATs. To simplify the simulations, we focused our analysis in GP130 and STAT1 proteins, two of the proteins that greatly vary in the different immune populations (***Figure 7a***). As baseline values for the concentrations $[GP130(0)]$, $[IL27R\alpha(0)]$, $[STAT1(0)]$ and $[STAT3(0)]$ we used approximately the median values from the posterior distributions for each parameter: $[GP130(0)] = 25\,\text{nM}$, $[IL27R\alpha(0)] = 50\,\text{nM}$ and $[STAT1(0)] = [STAT3(0)] = 500\,\text{nM}$. To see the effect of varying GP130 concentrations on pSTAT signaling, we decreased the initial concentration of GP130 and simulated the model using the accepted parameters sets from the ABC-SMC to inform the other parameter values. A tenfold reduction on GP130 concentration ($[GP130(0)] = 2.5\,\text{nM}$) resulted in a striking loss in pSTAT1 levels induced by HypIL-6, with very little effect on pSTAT3 levels induced by this cytokine (***Figure 7b***). pSTAT1/3 kinetics induced by IL-27, however, was not affected by this decrease in GP130 concentration (***Figure 7b***). Interestingly, the HypIL-6 signaling profile predicted by our model at low GP130 concentrations strongly resemble the one induced by HypIL-6 in Th-1 cells (***Figure 1c***), where very low levels of GP130 are found, further confirming the robustness of the predictions generated by our mathematical models. When the concentration of STAT1 was increased by a factor of ten $[STAT1(0)] = 5000\,\text{nM}$, both HypIL-6 and IL-27 induced significantly higher levels of pSTAT1 activation (***Figure 7b***). pSTAT3 levels were not affected for HypIL-6 stimulation but were decreased for IL-27 stimulation (***Figure 7b***), further indicating the competitive nature of the binding of STAT1 and STAT3 to IL-27Rα and GP130. Overall, our mathematical model predicts that changes on GP130 and STAT1 expression produce a substantial remodeling of the HypIL-6 and IL-27 signalosome, which ultimately could lead to aberrant responses.

## STAT1 protein levels in SLE patients modify HypIL-6 and IL-27 signaling responses

STAT1 is a classical IFN responsive gene and STAT1 levels are highly increased in environments rich in IFNs (***Wong et al., 1998***). Thus, we next ask whether STAT1 levels would be increased in SLE patients, an example of disease where IFNs have been shown to correlate with a poor prognosis, making use of available gene expression datasets (***Tokuyama et al., 2018***). We did not find differences in the expression of GP130, IL-6Rα or IL-27Rα in SLE patients (***Figure 7—figure supplement 2a***). However, we detected a considerable increase in the levels of STAT1 and STAT3 transcripts in these patients when compared to healthy controls, with the increase on STAT1 expression being

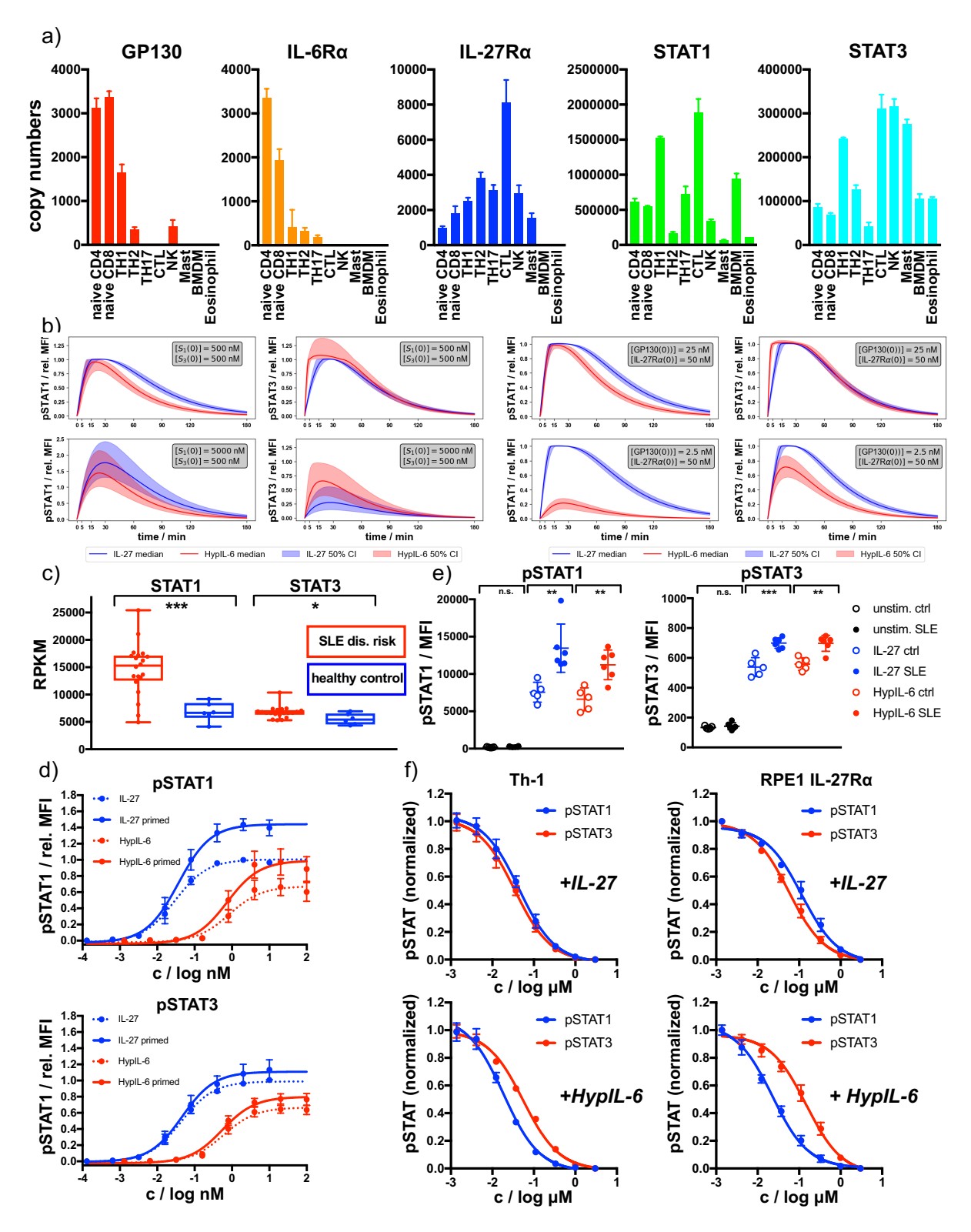

**Figure 7.** Receptor and STAT concentrations determine the nature of the cytokine response. (**a**) Copy numbers of indicated proteins determined for different T-cell subsets using mass-spectrometry-based proteomics (ImmPRes - http://immpres.co.uk). (**b**) Model predictions for varying levels of STAT1 and STAT3 (left panel) or IL-27Rα and GP130 (right panel) for phosphorylation kinetics of STAT1 and STAT3. (**c**) Gene expression profiles determined by RNAseq analysis comparing indicated genes of a cohort of SLE risk patients with a cohort of healthy controls. Data obtained from: *Proc Natl Acad Sci U*

*Figure 7 continued on next page*

*Figure 7 continued*

*S A* 115, 12565–12572. Two-tailed Student's T-test: *p<0.05, **p≤0.01,***p≤0.001; n.s., not significant. (**d**) Dose-dependent phosphorylation of STAT1 and STAT3 as a response to IL-27 and HypIL-6 stimulation in naive and IFNα2-primed (2 nM, 24 hr) Th-1 cells, normalized to maximal IL-27 stimulation (ctrl). Data was obtained from four biological replicates with each two technical replicates, showing mean ±std dev. (**e**) Phosphorylation of STAT1 (left) and STAT3 (right) as a response to IL-27 (2 nM, 15 min) and HypIL-6 (10 nM, 15 min) stimulation in healthy control (ctrl) and SLE patient CD4 +T cells. Data was obtained from five healthy control donors (*Diveu et al., 2009*) and six SLE patients. Two-tailed Student's T-test: *p<0.05, **p≤0.01, ***p≤0.001; n.s., not significant. (**f**) Tofacitinib titration to inhibit STAT1 and STAT3 phosphorylation by IL-27 (top, 2 nM, 15 min) HypIL-6 (bottom, 10 nM, 15 min) in Th-1 cells (left) and RPE1 IL-27Rα cells (right). Data was obtained from three donors with each two technical replicates (Th-1) and two biological experiments (RPE1 IL-27Rα cells) with each two technical replicates, showing mean ±std dev.

The online version of this article includes the following figure supplement(s) for figure 7:

**Figure supplement 1.** Validation of mathematical models for signaling kinetics of IL-27Rα receptor chimera and Y613F mutant.

**Figure supplement 2.** Altered total STAT levels and biased pSTAT1/pSTAT3 response in SLE.

significantly more pronounced (*Figure 7c*). Since our mathematical model predicted that increases in STAT1 expression could significantly change cytokine-induced cellular responses by HypIL-6 and IL-27, we next experimentally tested this prediction. For that, we primed Th-1 cells with IFNα2 over-night to increase total STAT1 levels (and to a lower extent STAT3) in these cells (*Figure 7—figure supplement 2b*). While both HypIL-6 and IL-27 induced comparable levels of pSTAT3 in primed and non-primed Th-1 cells, levels of pSTAT1 induced by the two cytokines were significantly upregulated in primed Th-1 cells, resulting in a bias STAT1 response and confirming our model predictions (*Figure 7d*). We next investigated whether this bias STAT1 activation by HypIL-6 and IL-27 observed in IFNα2-primed Th-1 cells was also present in SLE patients. For that we collected PBMCs from six SLE patients or five age-matched healthy controls and measured STAT1 and STAT3 expression, as well as pSTAT1 and pSTAT3 induction by HyIL-6 and IL-27 after 15 min treatments in CD4 T cells. Importantly, comparable results to those obtained with IFN-primed Th-1 cells were obtained, with signaling bias toward pSTAT1 in CD4 +T cells from SLE patients stimulated with HypIL-6 and IL-27 (*Figure 7e*, *Figure 7—figure supplement 2c & d*), further supporting the fact that STAT concentrations play a critical role in defining cytokine responses in autoimmune disorders.

Our data show that STAT1 and STAT3 compete for phospho-Tyr motifs in GP130, with STAT3 having an advantage resulting from its tighter affinity to GP130. Finally, we asked whether crippling JAK activity by using sub-saturating doses of JAK inhibitors could differentially affect STAT1 and STAT3 activation by HypIL-6 and therefore rescue the altered cytokine responses found in SLE patients. To test this, RPE1 and Th-1 cells were stimulated with saturated concentrations of HypIL-6 and titrating the concentrations of Tofacitinib, a clinically approved JAK inhibitor. Strikingly, Tofaciti-nib inhibited HypIL-6 induced pSTAT1 more efficiently than pSTAT3 in both RPE1 IL-27Rα cells and Th-1 cells (*Figure 7f*). At 50 nM concentration, Tofacitinib inhibited pSTAT1 levels induced by HypIL-6 by 60%, while only inhibited pSTAT3 levels by 30% (*Figure 7f*) – an effect that we did not observe for IL-27 stimulation. Overall, our results show that the changes in STATs concentration found in autoimmune disorders shape cytokine signaling responses and could contribute to disease progression.

## Discussion

Cytokine pleiotropy is the ability of a cytokine to exert a wide range of biological responses in different cell types. This functional pleiotropy has made the study of cytokine biology extremely challenging given the strong crosstalk and shared usage of key components of their signaling pathways, leading to a high degree of signaling plasticity, yet still allowing functional selectivity (*Garbers et al., 2012*; *Kang et al., 2020*). Here, we aimed to identify the underlying determinants that define cytokine functional selectivity by comparing IL-27 and IL-6 at multiple scales – ranging from cell surface receptors to proteomic changes. We show that IL-27 triggers a more sustained STAT1 phosphorylation than IL-6, via a high-affinity STAT1/IL-27Rα interaction centered around Tyr613 on IL-27Rα. This in turn results in a more sustained IRF1 expression induced by IL-27, which leads to the upregulation of a second wave of gene expression unique to IL-27 and comprised of classical ISGs. We go one step further and show that this strong receptor/STAT coupling is altered in autoimmune disorders where STATs concentrations are often dysregulated. Increased expression of

STAT1 in SLE patients' biases HypIL-6 and IL-27 responses toward STAT1 activation, further contributing to the worsening of the disease. By using suboptimal doses of the JAK inhibitor Tofacitinib, we show that specific STAT proteins engaged by a given cytokine can be targeted.

The tight coupling of one receptor subunit to one particular STAT that we have identified in our study is a rather unusual phenomenon for heterodimeric cytokine receptor complexes, which has been first suggested by *Owaki et al., 2008*. Generally, the entire signaling output driven by a cytokine-receptor complex emanates from a dominant receptor subunit, which carries several Tyr residues susceptible of being phosphorylated (*Umeshita-Suyama et al., 2000*; *Nadeau et al., 1999*). This in turn results in competition between different STATs for binding to shared phospho-Tyr motifs in the dominant receptor chain, leading to different kinetics of STAT phosphorylation as observed for IL-6 stimulation (*Rolvering et al., 2017*; *Figure 1b*). Moreover, this localized signaling quantum allows phosphatases and feedback regulators – induced upon cytokine stimulation – to act in synergy to reset the system to its basal state, generating a very synchronous and coordinated signaling wave. Although very effective, this molecular paradigm presents its limitations. STAT competition for the same pool of phospho-Tyr makes the system very sensitive to changes in STAT concentration. IFNγ-primed cells, which exhibit increased STAT1 levels, trigger an IFNγ-like STAT1 response upon IL-6 stimulation (*Costa-Pereira et al., 2002*). IL-10 anti-inflammatory properties are lost in cells with high levels of STAT1 expression, as a result of a pro-inflammatory environment rich in IFNs (*Sharif et al., 2004*). Indeed, we show that STAT1 transcripts levels are increased in Crohn's disease and SLE patients and they contributed to alter IL-6 responses. Strikingly, IL-27 appears to have evolved away from this general model of cytokine signaling activation. Our results show that STAT1 activation by IL-27 is tightly coupled to IL-27Rα, while STAT3 activation by this cytokine mostly depends on GP130. This decoupled STAT1 and STAT3 activation by IL-27 is possible thanks to the presence of a putative high-affinity STAT1 binding site on IL-27Rα that resembles the one present in IFNγR1 (*Pflanz et al., 2004*). As a result of this, IL-27 can trigger sustained and independent phosphorylation of both STAT1 and STAT3. This IL-27 feature allows it to induce robust responses in dynamic immune environments. Indeed, our mathematical models of cytokine signaling, and Bayesian inference, together with the experimental observations show that changes in receptor concentration minimally affected pSTAT1/3 induced by IL-27, while they fundamentally alter IL-6 responses. Overall, our data show that cytokine responses are versatile and adapt to the continuously changing cell proteome, highlighting the need to measure cytokine receptors and STATs expression levels, in addition to cytokine levels, in disease environments to better understand and predict altered responses elicited by dysregulated cytokines.

In recent years, it has become apparent that the stability of the cytokine-receptor complex influences signaling identity by cytokines (*Richter et al., 2017*). Short-lived complexes activate less efficiently those STAT molecules that bind with low-affinity phospho-Tyr motif in a given cytokine receptor (*Martinez-Fabregas et al., 2019*). Our current results further support this kinetic discrimination mechanism for STAT activation. Our statistical inference identified differences in STAT recognition to the cytokine receptor phospho-Tyr motifs as one of the major determinants of STAT phosphorylation kinetics. This parameter alone was sufficient to explain transient and sustained STAT1 phosphorylation induced by IL-6 and IL-27, respectively, without the need to invoke the action of phosphatases or negative feedback regulators such as SOCSs. Indeed, our results indicate that the rate of STAT1 dephosphorylation is similar between the IL-6 and IL-27 systems, suggesting that phosphatases do not contribute to these early kinetic differences. Moreover, blocking protein translation, and therefore the upregulation of negative feedback regulators by IL-6 treatment did not result in a more sustained STAT1 phosphorylation by IL-6, again indicating that the transient kinetics of STAT1 phosphorylation by IL-6 is encoded at the receptor level and does not require further regulation. However, recent reports have found that the amplitude of STAT1 phosphorylation in response to IL-6 is regulated by levels of PTPN2 expression, suggesting that phosphatases can play additional roles in shaping IL-6 responses beyond controlling the kinetics of STAT activation (*Twohig et al., 2019*). STAT1 phosphorylation levels by IL-27 on the other hand were significantly more sustained in the absence of protein translation, suggesting that negative feedback mechanisms are required to downmodulate signaling emanating from high-affinity STAT-receptor interactions. Overall, our results suggest that while phosphatases and negative feedback regulators play an important role in maintaining cytokine signaling homeostasis (*Heinrich et al., 2003*), the kinetics of

STAT activation appears to be already encoded at the level of receptor engagement, thus ensuring maximal efficiency and signal robustness.

Cytokine signaling plasticity can occur at the level of receptor activation. In the past years, a scenario has emerged suggesting that the absolute number of signaling active receptor complexes is a critical determinant for signal output integration. Accordingly, specific biological responses were shown to be tuned either by abundance of cell surface receptors (*Levin et al., 2011*; *Moraga et al., 2009*) or by the level of receptor assembly (*Martinez-Fabregas et al., 2019*; *Wilmes et al., 2015*; *Ho et al., 2017*). Here, we show for the first time that IL-27-induced dimerization of IL-27Rα and GP130 at the cell surface of live cells – in good agreement with previous studies on heterodimeric cytokine receptor systems (*Wilmes et al., 2015*; *Richter et al., 2017*). For IL-27, the receptor subunits IL-27Rα and GP130 can be expressed at different ratios as seen for naive vs. activated T-cells (*Charlot-Rabiega et al., 2011*) as well as intestinal cells (*Diegelmann et al., 2012*). On T-cells, particularly after activation, IL-27Rα is expressed in strong excess over GP130, rendering GP130 as the limiting factor for receptor complex assembly (*Pflanz et al., 2004*). Interestingly, we observe that in addition to a faster kinetic of STAT1 phosphorylation, HypIL-6 treatment induces a lower maximal amplitude in pSTAT1 activation in T cells. This is in stark contrast to our results in RPE1 cells, where high abundance of GP130 (~3000–4000 copies of cell surface GP130) is found. In these cells, both cytokines elicited similar amplitudes of STAT1 phosphorylation. Our results suggest that surface receptor density in synergy with STATs binding dynamics to phospho-Tyr motif on cytokine receptors act to define the amplitude and kinetics of STAT activation in response to cytokine stimulation.

The distinct STAT1 and STAT3 kinetic profiles induced by IL-6 and IL-27 are the prerequisite for time-correlated decoupling of genetic programs: a 'shared GP130/STAT3-dependent wave' and an IL-27-'unique IL-27Rα/STAT1-dependent wave'. However, pSTAT1 levels induced by IL-27 at 3 hr were down to ~10% of maximal amplitude, suggesting that additional factors would be required to amplify the initial STAT1 response elicited by IL-27. We observed that IL-27 induces the expression of an early wave of classical STAT1-dependent genes, which is also shared by IL-6. However, while IL-27 induces the upregulation of these genes throughout the entire duration of the experiment, IL-6 only resulted in a transient spike. We reasoned that this additional factor required for IL-27 signal amplification would be among these early STAT1-dependent genes. Among this set of genes we found the transcription factor IRF1, which had been shown to act as a feedback amplificant for pSTAT1 activity (*Zenke et al., 2018*). Importantly, IRF1 protein levels have been shown to be upregulated in response to IL-27 and IFNγ but not to IL-6 stimulation in hepatocytes (*Bender et al., 2009*). IRF1 plays a key role in chromatin accessibility which is critically required for IL-27-induced differentiation of Tr-1 cells and subsequent IL-10 secretion (*Karwacz et al., 2017*). Here, we could prove that the contribution of IRF1 on STAT1- but not STAT3-dependent genes is a generic feature of IL-27 signaling. This readily explains the significant transcriptomic overlap of IL-27 with type I (*Imamichi et al., 2012*) or type II interferons (*Rolvering et al., 2017*) after long-term stimulation with these cytokines. Along this line, it is not surprising that IL-27 – beyond its well-described effects on T-cell development – can also mount a considerable antiviral response as shown in hepatic cells and PBMCs (*Fakruddin et al., 2007*; *Frank et al., 2010*). Our results suggest that by modulating the kinetics of STAT phosphorylation, cytokines can modulate the expression of accessory transcription factors, such as IRF1, that act in synergy with STATs to fine-tune gene expression and provide functional diversity.

## Data availability

Python (version 3.7) codes for the ABC-SMC model selection and parameter inference can be found in the public repository https://github.com/PollyJeffrey/Cytokine_modelling (copy archived at swh:1:rev:9c3e0ddc7a96eac941baad560d1541d660b0515d; *Wilmes, 2021*), along with the results of the analysis. Phospho-proteomic and proteomic datasets were uploaded to the Proteome Exchange platform with accession numbers PXD024657 and PXD024188 respectively. RNA-seq dataset was uploaded in the GSE database with accession number GSE164479.

## Materials and methods

### Protein expression and purification

Murine IL-27 was cloned as a linker-connected single-chain variant (p28 +EBI3) as described in *Oniki et al., 2006*. Human HyperIL-6 (HypIL-6), and murine single-chain IL-27 were cloned into the pAcGP67-A vector (BD Biosciences) in frame with an N-terminal gp67 signal sequence and a C-terminal hexahistidine tag, and produced using the baculovirus expression system, as described in *LaPorte et al., 2008*. Baculovirus stocks were prepared by transfection and amplification in *Spodoptera frugiperda* (*Sf9*) cells grown in SF900II media (Invitrogen) and protein expression was carried out in suspension *Trichoplusiani ni* (High Five) cells grown in InsectXpress media (Lonza).

Purification was performed using the method described in *Spangler et al., 2019*. For IL-27, the cells were pelleted with centrifugation at 2000 rpm, prior to a precipitation step through addition of Tris pH 8.0, $CaCl_2$ and $NiCl_2$ to final concentrations of 200 mM, 50 mM and 1 mM, respectively. The precipitate formed was then removed through centrifugation at 6000 rpm. Nickel-NTA agarose beads (Qiagen) were added and the target proteins purified through batch binding followed by column washing in HBS-Hi buffer (HBS buffer supplemented to 500 mM NaCl and 5% glycerol, pH 7.2). Elution was performed using HBS-Hi buffer plus 200 mM imidazole. Final purification was performed by size exclusion chromatography on an ENrich SEC 650 300 column (Biorad), again equilibrated in HBS-Hi. Concentration of the purified sample was carried out using 10 kDa Millipore Amicon-Ultra spin concentrators. For HypIL-6, proteins were purified likewise, but in 10 mM HEPES (pH 7.2) containing 150 mM NaCl. Recombinant cytokines were purified to greater than 98% homogeneity.

For cell surface labeling, the anti-GFP nanobody (NB) 'enhancer' and 'minimizer' were used, which bind mEGFP with subnanomolar binding affinity (*Kirchhofer et al., 2010*). NB was cloned into pET-21a with an additional cysteine at the C-terminus for site-specific fluorophore conjugation in a 1:1 fluorophore:nanobody stoichiometry. Furthermore, (PAS)$_5$ sequence to increase protein stability and a His-tag for purification were fused at the C-terminus. Protein expression in *E. coli* Rosetta (DE3) and purification by immobilized metal ion affinity chromatography was carried out by standard protocols. Purified protein was dialyzed against HEPES pH 7.5 and reacted with a twofold molar excess of DY647 maleimide (Dyomics), ATTO 643 maleimide (AT643) and ATTO Rho11 maleimide (Rho11) (ATTO-TEC GmbH), respectively. After 1 hr, a threefold molar excess (with respect to the maleimide) of cysteine was added to quench excess dye. Protein aggregates and free dye were subsequently removed by size exclusion chromatography (SEC). A labeling degree of 0.9-1:1 fluorophore:protein was achieved as determined by UV/Vis spectrophotometry.

### CD4 +T cell purification and Th-1 differentiation

Human buffy coats were obtained from the Scottish Blood Transfusion Service and peripheral blood mononuclear cells (PBMCs) of healthy donors were isolated from buffy coat samples by density gradient centrifugation according to manufacturer's protocols (Lymphoprep, STEMCELL Technologies). From each donor, $100 \times 10^6$ PBMCs were used for isolation of CD4 +T cells. Cells were decorated with anti-CD4$^{FITC}$ antibodies (Biolegend, #357406) and isolated by magnetic separation according to manufacturer's protocols (MACS Miltenyi) to a purity >98% CD4+. Freshly isolated resting CD4$^+$ T cells ($3 \times 10^7$ per donor) were activated under Th-1 polarizing conditions using ImmunoCult Human CD3/CD28 T Cell Activator (StemCell, Cat#10971) following manufacturer instructions for 3 days in RPMI-1640, 10% v/v FBS, 100 U/ml penicillin-streptomycin (Gibco) in the presence of the cytokines IL-2 (Novartis, #709421, 20 ng/ml), anti-IL-4 antibody (10 ng/ml, BD Biosciences, #554481), IL-12 (20 ng/ml, BioLegend, #573002). After 3 days of priming, cells were expanded for another 5 days in the presence of IL-2 (20 ng/ml).

### Human SLE patient samples

This study was authorized by the French Competent Authority dealing with Research on Human Biological Samples namely the French Ministry of Research. The Authorization number is ECH 19/04. To issue such authorization, the Ministry of Research has sought the advice of an independent ethics committee, namely the 'Comité de Protection des Personnes,' which voted positively, and all patients gave their written informed consent. The healthy volunteer was recruited to serve as healthy control individuals. Healthy and patients' blood samples were collected in heparinized tubes (BD

Vacutainer 368886, BD Biosciences San Jose, CA, USA) and PBMC samples were isolated using Ficoll (Pancoll, Pan Biotech #P04-60500) density gradient centrifugation. The isolated PBMCs were washed with PBS and the remaining red blood cells were lysed using RBC lysis buffer (ACK lysing buffer, Gibco #A10492-01), incubate 3 min at room temperature. Cells were washed in PBS and resuspend the cells with 1 ml of freezing medium (with DMSO, PAN Biotech, #P07-90050) and transfer the cells in a cryotube. cryotube in a Freezing container (Nalgene) and at −80°C and then transferred into liquid nitrogen container for long-term storage.

## Classification and demographic information about SLE patients and healthy controls

SLE patients were included if they fulfilled the American College of Rheumatology (ACR) Classification Criteria Hochberg MC. Updating the American College of Rheumatology revised criteria for the classification of systemic lupus erythematosus (*Hochberg, 1997*). Exclusion criteria were current intake of 10 mg or more of prednisone or equivalent and/or use of immunosuppressants within the previous 6 months before inclusion. Use of hydroxychloroquine was not an exclusion criterion. Patients were mostly in clinical remission, half with biological remission, half with persistent anti native DNA autoantibodies. All SLE patients and healthy controls were females between 41 and 58 years old.

## (Phospho-) Proteomics

For (phospho-) proteomic experiments, Th-1 cells from each donor were split into three different conditions after initial expansion: Light SILAC media (40 mg/ml L-Lysine K0 (Sigma, #L8662) and 84 mg/ml L-Arginine R0 (Sigma, #A8094)), medium SILAC media (49 mg/ml L-Lysine U-13C6 K6 (CKGAS, #CLM-2247–0.25) and 103 mg/ml L-Arginine U-13C6 R6 (CKGAS, #CLM-2265–0.25)) and heavy SILAC media (49.7 mg/ml L-Lysine U-13C6,U-15N2 K8 (CKGAS, #CNLM-291-H-0.25) and 105.8 mg/ml L-Arginine U-13C6,U-15N2 R10 (CKGAS, #CNLM-539-H-0.25)) prepared in RPMI SILAC media (Thermo Scientific, #88365) supplemented with 10% dialyzed FBS (HyClone, #SH30079.03), 5 ml L-Glutamine (Invitrogen, #25030024), 5 ml Pen/Strep (Invitrogen, #15140122), 5 ml MEM vitamin solution (Thermo Scientific, #11120052), 5 ml Selenium-Transferrin-Insulin (Thermo Scientific, #41400045) and expanded in the presence of 20 ng/ml IL-2 and 10 ng/ml anti-IL4 for another 10 days in order to achieve complete labeling. Media was exchanged every 2 days. Incorporation of medium and heavy version of Lysine and Arginine was checked by mass spectrometry and samples with an incorporation greater than 95% were used.

After expansion, cells were starved without IL-2 for 24 hr before stimulation with 10 nM IL-27 or 20 nM HyIL-6 for 15 min (phosphoproteomics) or 24 hr (global proteomic changes). Cells were then washed three times in ice-cold PBS, mix in a 1:1:1 ratio, resuspended in SDS-containing lysis buffer (1% SDS in 100 mM Triethylammonium Bicarbonate buffer (TEAB)) and incubated on ice for 10 min to ensure cell lysis. Then, cell lysates were centrifuged at 20,000 g for 10 min at +4°C and supernatant was transferred to a clean tube. Protein concentration was determined by using BCA Protein Assay Kit (Thermo, #23227), and 10 mg of protein per experiment were reduced with 10 mM dithiothreitol (DTT, Sigma, #D0632) for 1 hr at 55°C and alkylated with 20 mM iodoacetamide (IAA, Sigma, #I6125) for 30 min at RT. Protein was then precipitated using six volumes of chilled (−20°C) acetone overnight. After precipitation, protein pellet was resuspended in 1 ml of 100 mM TEAB and digested with Trypsin (1:100 w/w, Thermo, #90058) and digested overnight at 37°C. Then, samples were cleared by centrifugation at 20000 g for 30 min at +4°C, and peptide concentration was quantified with Quantitative Colorimetric Peptide Assay (Thermo, #23275).

Phosphopeptide enrichment in the peptide fractions generated as described above was carried out using MagResyn Ti-IMAC following manufacturer instructions (2BScientific, MRTIM002).

## High pH reverse phase fractionation for phosphoproteomics

Samples were dissolved in 200 µL of 10 mM ammonium formate buffer pH 9.5 and peptides are fractionated using high pH RP chromatography. A C18 Column from Waters (XBridge peptide BEH, 130 Å, 3.5 µm 4.6 × 150 mm, Ireland) with a guard column (XBridge, C18, 3.5 µm, 4.6 × 20 mm, Waters) are used on an Ultimate 3000 HPLC (Thermo-Scientific). Buffers A and B used for fractionation consist, respectively of 10 mM ammonium formate in milliQ water (Buffer A) and 10 mM ammonium

formate in 90% acetonitrile (Buffer B), both buffers were adjusted to pH 9.5 with ammonia. Fractions are collected using a WPS-3000FC autosampler (Thermo-Scientific) at 1 min intervals. Column and guard column were equilibrated with 2% buffer B for 20 min at a constant flow rate of 0.8 ml/min and a constant temperature 0 f 21°C. Samples (193 µl) are loaded onto the column at 0.8 ml/min, and separation gradient started from 2% buffer B, to 8% B in 6 min, then from 8% B to 45% B within 54 min and finally from 45% B to 100% B in 5 min. The column is washed for 15 min at 100% buffer B and equilibrated at 2% buffer B for 20 min as mentioned above. The fraction collection started 1 min after injection and stopped after 80 min (total of 80 fractions, 800 µl each). Each peptide fraction was acidified immediately after elution from the column by adding 20–30 µl 10% formic acid to each tube in the autosampler. The total number of fractions concatenated was set to 10. The content of fractions from each set was dried prior to further analysis.

## LC-MS/MS analysis

LC-MS analysis was done at the FingerPrints Proteomics Facility (University of Dundee). Analysis of peptide readout was performed on a Q Exactive plus, Mass Spectrometer (Thermo Scientific) coupled with a Dionex Ultimate 3000 RS (Thermo Scientific). LC buffers used are the following: buffer A (0.1% formic acid in Milli-Q water (v/v)) and buffer B 80% acetonitrile and 0.1% formic acid in Milli-Q water (v/v). Dried fractions were resuspended in 35 µl, 1% formic acid and aliquots of 15 µl of each fraction were loaded at 10 µl/min onto a trap column (100 µm × 2 cm, PepMap nanoViper C18 column, 5 µm, 100 Å, Thermo Scientific) equilibrated in 0.1% TFA. The trap column was washed for 5 min at the same flow rate with 0.1% TFA and then switched in-line with a Thermo Scientific, resolving C18 column (75 µm × 50 cm, PepMap RSLC C18 column, 2 µm, 100 Å). The peptides were eluted from the column at a constant flow rate of 300 nl/min with a linear gradient from 2% buffer B to 5% buffer B in 5 min then from 5% buffer B to 35% buffer B in 125 min, and finally from 35% buffer B to 98% buffer B in 2 min. The column was then washed with 98% buffer B for 20 min and re-equilibrated in 2% buffer B for 17 min. The column was kept at a constant temperature of 50°C. Q-exactive plus was operated in data dependent positive ionization mode. The source voltage was set to 2.5 Kv and the capillary temperature was 250°C.

A scan cycle comprised MS1 scan (m/z range from 350 to 1600, ion injection time of 20 ms, resolution 70,000 and automatic gain control (AGC) $1 \times 10^6$) acquired in profile mode, followed by 15 sequential dependent MS2 scans (resolution 17500) of the most intense ions fulfilling predefined selection criteria (AGC $2 \times 10^5$, maximum ion injection time 100 ms, isolation window of 1.4 m/z, fixed first mass of 100 m/z, spectrum data type: centroid, intensity threshold $2 \times 10^4$, exclusion of unassigned, singly and >7 charged precursors, peptide match preferred, exclude isotopes on, dynamic exclusion time 45 s). The HCD collision energy was set to 27% of the normalized collision energy. Mass accuracy is checked before the start of samples analysis.

## Mass spectrometry data analysis

Q Exactive Plus Mass Spectrometer. RAW files were analyzed, and peptides and proteins quantified using MaxQuant (*Cox and Mann, 2008*), using the built-in search engine Andromeda (*Cox et al., 2011*). All settings were set as default, except for the minimal peptide length of 5, and Andromeda search engine was configured for the UniProt *Homo sapiens* protein database (release date: 2018_09). Peptide and protein ratios only quantified in at least two out of the three replicates were considered, and the p-values were determined by Student's t test and corrected for multiple testing using the Benjamini–Hochberg procedure (Benjamini and Hochberg, 1995).

## Plasmid constructs

For single-molecule fluorescence microscopy, monomeric non-fluorescent (Y67F) variant of eGFP was N-terminally fused to GP130. This tag (mXFP**m**) was engineered to specifically bind anti-GFP nanobody 'minimizer' (αGFP-miNB). This construct was inserted into a modified version of pSems-26 m (Covalys) using a signal peptide of Igk. The ORF was linked to a neomycin resistance cassette via an IRES site. A mXFPe-IL-27Rα construct was designed likewise but is recognized by αGFP nanobody 'enhancer' (mXFP**e**). The chimeric construct mXFP-IL-27Rα (ECD and TMD)-GP130(ICD) was a fusion construct of IL-27Rα (aa 33–540) and GP130 (aa 645–918).

## Cell lines and media

RPE1 cells were grown in DMEM/F12 containing 10% v/v FBS, penicillin-streptomycin, and L-glutamine (2 mM). RPE1 cells were stably transfected by mXFPe-IL-27Rα ( = 'RPE1 IL-27Rα"), mutants and the chimeric construct by PEI method according to standard protocols. Using G418 selection (0.6 mg/ml) individual clones were selected, proliferated and characterized. RPE1 cells previously knocked out for endogenous GP130 (*Martinez-Fabregas et al., 2019*) were stably transfected as above with mXFPm-GP130 ( = 'RPE1 10x[GP130]").

## Flow cytometry staining and antibodies

For measuring dose-response curves of STAT1/3 phosphorylation (either Th-1 cells or RPE1 clones), 96-well plated were prepared with 50 µl of cell suspensions at $2 \times 10^6$ cells/ml/well for Th-1 and $2 \times 10^5$ cells/ml/well for RPE1. The latter were detached using Accutase (Sigma). Cells were stimulated with a set of different concentrations to obtain dose-response curves. To this end, cells were stimulated for 15 min at 37°C with the respective cytokines followed by PFA fixation (2%) for 15 min at RT.

For kinetic experiments, cell suspensions were stimulated with a defined, saturating concentration of cytokines (10 nM IL-27, 20 nM HypIL-6, 100 nM wt-IL-6) in a reverse order so that all cell suspensions were PFA-fixed (2%) simultaneously. For pSTAT1/3 kinetic experiments at JAK inhibition, Tofacitinib (2 µM, Stratech, #S2789-SEL) was added after 15 min of stimulation and cells were PFA-fixed in correct order.

After fixation (15 min at RT), cells were spun down at 300 g for 6 min at 4°C. Cell pellets were resuspended and permeabilized in ice-cold methanol and kept for 30 min on ice. After permeabilization cells were fluorescently barcoded according to *Krutzik and Nolan, 2006*. In brief: using two NHS-dyes (PacificBlue, #10163, DyLight800, #46421, Thermo Scientific), individual wells were stained with a combination of different concentrations of these dyes. After barcoding, cells are pooled and stained with anti-pSTAT1[Alexa647] (Cell Signaling Technologies, #8009) and anti-pSTAT3[Alexa488] (Biolegend, #651006) at a 1:100 dilution in PBS + 0.5% BSA for 1 hr at RT. T-cells were also stained with anti-CD8[AlexaFlour700] (1:120, Biolegend, #300920), anti-CD4[PE] (1:120, Biolegend, #357404), anti-CD3[BrilliantViolet510] (1:100, Biolegend, #300448). Cells were analzyed at the flow cytometer (Beckman Coulter, Cytoflex S) and individual cell populations were identified by their barcoding pattern (supp. Fig 1b). Mean fluorescence intensity (MFI) of pSTAT1[647] and pSTAT3[488] was measured for all individual cell populations.

For measuring total STAT levels, methanol-permeabilized cells were stained with anti-STAT1[Alexa647] (1:70, Biolegend, #558560) or anti-STAT3[APC] (1:50, Biolegend, #560392). Total IRF1 levels methanol-permeabilized cells were stained with anti-IRF1[Alexa647] (1:50, Biolegend, #14105). For measuring cell surface levels of GP130, cells were detached with Accutase (Sigma) and stained with anti-GP130[APC] (1:100, Biolegend, #362006) for 1 hr on ice. For measuring cell surface levels of mXFPe-IL-27Rα variants, cells were detached with Accutase (Sigma) and stained with αGFP-enNB[Dy647] (10 nM) for 30 min on ice.

## RNA transcriptome sequencing

Human Th-1 cells from three donors each (StemCell Technologies) were cultivated and stimulated as described in above. Cells were washed in Hank's balanced salt solution (HBSS, Gibco) and snap frozen for storage. RNA was isolated using the RNeasy Kit (Quiagen) according to manufacturer's protocol. All RNA 260/280 ratios were above 1.9. Of each sample, 1 µg of RNA was used. Transcriptomic analysis was done by Novogene as follows. Sequencing libraries were generated using NEBNext UltraTM RNALibrary Prep Kit for Illumina (NEB, USA) following manufacturer's recommendations and index codes were added to attribute sequences to each sample. Briefly, mRNA was purified from total RNA using poly-T oligo-attached magnetic beads. Fragmentation was carried out using divalent cations under elevated temperature in NEBNext First StrandSynthesis Reaction Buffer (5X). First strand cDNA was synthesized using random hexamer primer and M-MuLV Reverse Transcriptase (RNase H-). Second strand cDNA synthesis was subsequently performed using DNA Polymerase I and RNase H. Remaining overhangs were converted into blunt ends via exonuclease/polymerase activities. After adenylation of 3' ends of DNA fragments, NEBNext Adaptor with hairpin loop structure were ligated to prepare for hybridization. In order to select cDNA fragments of preferentially 150 ~ 200 bp in length, the library fragments were purified with AMPure XP system

(Beckman Coulter, Beverly, USA). Then 3 µl USER Enzyme (NEB, USA) was used with size-selected, adaptor-ligated cDNA at 37℃ for 15 min followed by 5 min at 95℃ before PCR. Then PCR was performed with Phusion High-Fidelity DNA polymerase, Universal PCR primers and Index (X) Primer. At last, PCR products were purified (AMPure XP system) and library quality was assessed on the Agilent Bioanalyzer 2100 system.

## RNA sequencing data analysis

Primary data analysis for quality control, mapping to reference genome and quantification was conducted by Novogene as outlined below.

Quality control: Raw data (raw reads) of FASTQ format were firstly processed through in-house scripts. In this step, clean data (clean reads) were obtained by removing reads containing adapter and poly-N sequences and reads with low quality from raw data. At the same time, Q20, Q30 and GC content of the clean data were calculated. All the downstream analyses were based on the clean data with high quality.

Mapping to reference genome: Reference genome and gene model annotation files were downloaded from genome website browser (NCBI/UCSC/Ensembl) directly. Paired-end clean reads were mapped to the reference genome using HISAT2 software. HISAT2 uses a large set of small GFM indexes that collectively cover the whole genome. These small indexes (called local indexes), combined with several alignment strategies, enable rapid and accurate alignment of sequencing reads.

Quantification: HTSeq was used to count the read numbers mapped of each gene, including known and novel genes. And then RPKM of each gene was calculated based on the length of the gene and reads count mapped to this gene. RPKM, (Reads Per Kilobase of exon model per Million mapped reads), considers the effect of sequencing depth and gene length for the reads count at the same time and is currently the most commonly used method for estimating gene expression levels.

For each identified gene, the fold change was calculated by the ratio of cytokine stimulated/unstimulated expression levels within each donor and an unpaired, two-tailed t test was applied to calculate p values. Genes were considered to be significantly altered if: p value $\leq$ 0.05, and $\log_2$ fold change $\geq +1$ or $\leq -1$. Genes with an RPKM of less than one in two or more donors were excluded from analysis so as to remove genes with abundance near detection limit. Genes without annotated function were also removed. Functional annotation of genes (KEGG pathways, GO terms) was done using DAVID Bioinformatics Resource functional annotation tool (*Huang et al., 2009a*; *Huang et al., 2009b*). Clustered heatmap was generated using R Studio Pheatmap package.

## siRNA-mediated knockdown of IRF1 in RPE1 cells

A set of four IRF1-siRNAs were purchased from Dharmacon and tested individually to determine levels of knockdown achieved. The siRNA providing the highest level of IRF1. knockdown (Horizon, LQ-011704-00-0005, siRNA #2: UGAACUCCCUGCCAGAUAU) were subsequently used in all the experiments. RPE1 IL-27Rα cells were plated in six-well dishes (0.4 × $10^6$ cells per well) and transfected the next day with IRF1-siRNA or control-GAPDH siRNA (Horizon, D-001830-10-05) (Dharmacon) using DharmaFect one transfection reagent (Dharmacon) following the manufacturer's instructions for 24 hr. At different timepoints of IL-27 (2 nM) or HypIL-6 (10 nM) stimulation, samples were collected from each one six-well. Cells were trypsinized and each sample was spun down and pellets snap-frozen in liquid nitrogen for subsequent RNA isolation (90%) or PFA-fixed for total IRF1 staining (10%) by flow cytometry.

## Real-time quantitative PCR

Cells were subject to RNA isolation using the Qiagen RNeasy kit. RNA (100 ng) was reverse transcribed to complementary DNA (cDNA) using an iScript cDNA synthesis kit (BioRad, #1708890), which was used as template for quantitative PCR. PowerTrack SYBR Green Master Mix (Takara, #A46109) was used for the reaction with the following primers:

| Target | For | Rev | Size |
|---|---|---|---|
| β-actin | CATGTACGTTGCTATCCAGGC | CTCCTTAATGTCACGCACGAT | 250 bp |

*Continued on next page*

*Continued*

| Target | For | Rev | Size |
|--------|-----|-----|------|
| STAT1 | CTAGTGGAGTGGAAGCGGAG | CACCACAAACGAGCTCTGAA | 252 bp |
| GBP5 | TCCTCGGATTATTGCTCGGC | CCTTTGCGCTTCAGCCTTTT | 309 bp |
| OAS1 | GAAGGCAGCTCACGAAACC | AGGCCTCAGCCTCTTGTG | 114 bp |
| SOCS3 | GTCCCCCCAGAAGAGCCTATTA | TTGACGGTCTTCCGACAGAGAT | 118 |

β-actin was used as housekeeping gene for normalization. Each siRNA knockdown experiment was performed in four replicates with each sample for qPCR being done in two technical replicates.

## Mathematical models

We developed two new mathematical models, making use of ordinary differential equations (ODEs), for the initial steps of cytokine-receptor binding, dimer formation and signal activation by HypIL-6 and IL-27, respectively; namely, a set of ODEs for the HypIL-6 system and a separate one for the IL-27 system (see Supplementary material 1 for the set of ODEs describing each model). These ODEs describe the rate of change of the concentration for each molecular species considered in the receptor-ligand systems (HypIL-6 and IL-27) over time. By solving these ODEs, a time-course for the concentration of total (free and bound) phosphorylated STAT1 and STAT3 can be obtained and compared to the experimental data (Supp. Fig. 5c and 5d). The HypIL-6 and IL-27 mathematical models differ due to the reactions involved in the formation of the signaling dimer for each cytokine. Under stimulation with HypIL-6, two HypIL-6 bound GP130 monomers are required to form the homodimer (Supp. Fig. 4b), *Figure 2—figure supplement 1. Schematic model of involved reactions and parameters for IL-27 and HypIL-6 receptor activation.* whereas under IL-27 stimulation, we assume that IL-27 binds to the IL-27Rα chain and not to GP130 (Supp. Fig. 4c) and hence the heterodimer is comprised of an IL-27 molecule bound to an IL-27Rα monomer and one GP130 chain. In the mathematical models, we assume that upon formation of the dimers (homo- or heterodimer), these receptor chains become immediately phosphorylated. The models do not consider JAK molecules explicitly. We are assuming that these molecules are constitutively bound to their corresponding receptor chains and that they phosphorylate immediately upon receptor phosphorylation (dimer formation). After the formation of the dimer, which we denote by $D_6$ or $D_{27}$, formed by HypIL-6 or IL-27 respectively, the biochemical reactions included in each mathematical model are similar, and are summarized as follows. *Table 2* provides a description of the rates for each reaction considered in each (and both) mathematical model(s). In what follows we assume mass action kinetics for all the reactions. A free cytoplasmic unphosphorylated STAT1 or STAT3 molecule can bind to either receptor chain in the dimer, provided that the intracellular tyrosine residue of the receptor in the dimer is free (Supp. Fig. 4d and 4e). The STAT1 or STAT3 molecule can subsequently dissociate from the receptor chain in the dimer or can become phosphorylated (with rate $q$) whilst bound to the dimer. We have assumed that the rate of STAT1 or STAT3 phosphorylation when bound does not depend on the STAT type (1 or 3) or on the receptor chain (Supp. Fig. 4d and 4e). Phosphorylated STAT1 (pSTAT1) and STAT3 (pSTAT3) molecules can dissociate from the dimer. Once free in the cytoplasm, they can then dephosphorylate (Supp. Fig. 4h). We have assumed that this rate of STAT dephosphorylation only depends on the concentration of the respective pSTAT type. We note that no allostery has been considered in the models and hence, phosphorylated and unphosphorylated STAT molecules dissociate from the receptor with the same rate (Supp. Fig. 4d and 4e). Finally, any molecular species containing receptor molecules can be lost from the system, due to internalization or degradation processes, via one of two hypothesized mechanisms (Supp. Fig. 4f and 4g):

- hypothesis 1 ($H_1$): receptors (free or bound, phosphorylated or unphosphorylated) are internalized/degraded with a rate proportional to the concentration of the species in which they are contained, or
- hypothesis 2 ($H_2$): receptors (free or bound, phosphorylated or unphosphorylated) are internalized/degraded with a rate proportional to the product of the concentration of the species in which they are contained and the sum of the concentrations of free cytoplasmic phosphorylated STAT1 and STAT3.

**Table 2.** Parameter mathematical notation, description, and units, where $i \in \{1, 3\}$ so that STAT $i$ corresponds to STAT1 or STAT3.

| Parameter | Description | Unit |
|---|---|---|
| $r_{1,6}^{+}, r_{1,27}^{+}$ | Rate of receptor-ligand binding | nM$^{-1}$s$^{-1}$ |
| $r_{1,6}^{-}, r_{1,27}^{-}$ | Rate of receptor-ligand dissociation | s$^{-1}$ |
| $r_{2,6}^{+}, r_{2,27}^{+}$ | Rate of monomer-monomer binding (or dimer formation) | nM$^{-1}$s$^{-1}$ |
| $r_{2,6}^{-}, r_{2,27}^{-}$ | Rate of dimer dissociation | s$^{-1}$ |
| $k_{ia}^{+}$ | Rate of STATi binding to GP130 | nM$^{-1}$s$^{-1}$ |
| $k_{ib}^{+}$ | Rate of STATi binding to IL-27Rα | nM$^{-1}$s$^{-1}$ |
| $k_{ia}^{-}$ | Rate of STATi dissociating GP130 | s$^{-1}$ |
| $k_{ib}^{-}$ | Rate of STATi dissociating IL-27Rα | s$^{-1}$ |
| $q$ | Rate of STATi phosphorylation on a dimer | s$^{-1}$ |
| $d_i$ | Rate of pSTATi dephosphorylation | s$^{-1}$ |
| $\beta_6, \beta_{27}$ | Rate of receptor internalization/degradation under hypothesis 1 | s$^{-1}$ |
| $\gamma_6, \gamma_{27}$ | Rate of receptor internalization/degradation under hypothesis 2 | nM$^{-1}$s$^{-1}$ |
| $[R_1(0)]$ | Initial concentration of GP130 | nM |
| $[R_2(0)]$ | Initial concentration of IL-27Rα | nM |
| $[S_i(0)]$ | Initial concentration of STATi | nM |

We note that hypothesis 1 assumes that receptor molecules (free or bound, phosphorylated or unphosphorylated) are being internalized/degraded as part of the natural cellular trafficking cycle. Hypothesis 2 is consistent with a potential feedback mechanism, whereby the free cytoplasmic pSTAT molecules would migrate to the nucleus and increase the translation of negative feedback proteins, such as SOCS3, which down-regulate cytokine signaling. Thus, the internalization/degradation rate of receptor molecules (free or bound, phosphorylated or unphosphorylated) under hypothesis 2 increases with the total amount of free cytoplasmic phosphorylated STAT1 and STAT3, to account for this negative feedback on surface receptor expression. A depiction of the reactions in both the HypIL-6 and IL-27 mathematical models and under each hypothesis is given in Supp. Fig. 4 where (b), (d), (f) and (h) describe the HypIL-6 model and (c), (e), (g) and (h) describe the IL-27 model. In this figure, $i \in \{1, 3\}$ so that the reactions shown can involve STAT1 or STAT3. Each reaction arrow has been shown with its rate (above or below the arrow). We note that we assume mass action kinetics for each reaction. The notation for the rate constants and initial molecular concentrations in the models, along with their descriptions and units, are given in *Table 2*.

The HypIL-6 mathematical model was formulated based on biochemical reactions involving the following species:

- $L_6 =$ HypIL-6,
- $R_1 =$ GP130,
- $C_1 =$ GP130 - HypIL-6 monomer,
- $D_6 =$ phosphorylated GP130 - HypIL-6 - HypIL-6 - GP130 homodimer,
- $S_1 =$ unbound cytoplasmic unphosphorylated STAT1,
- $S_3 =$ unbound cytoplasmic unphosphorylated STAT3,
- $D_6 \cdot S_1 =$ dimer bound to STAT1,
- $D_6 \cdot S_3 =$ dimer bound to STAT3,
- $D_6 \cdot pS_1 =$ dimer bound to pSTAT1,
- $D_6 \cdot pS_3 =$ dimer bound to pSTAT3,
- $S_1 \cdot D_6 \cdot S_1 =$ dimer bound to two molecules of STAT1,
- $pS_1 \cdot D_6 \cdot S_1 =$ dimer bound to two molecules of STAT1, one of which is phosphorylated,
- $pS_1 \cdot D_6 \cdot pS_1 =$ dimer bound to two molecules of pSTAT1,
- $S_3 \cdot D_6 \cdot S_3 =$ dimer bound to two molecules of STAT3,
- $pS_3 \cdot D_6 \cdot S_3 =$ dimer bound to two molecules of STAT3, one of which is phosphorylated,
- $pS_3 \cdot D_6 \cdot pS_3 =$ dimer bound to two molecules of pSTAT3,

- $S_1 \cdot D_6 \cdot S_3 =$ dimer bound to one molecule of STAT1 and one of STAT3,
- $pS_1 \cdot D_6 \cdot S_3 =$ dimer bound to one molecule of pSTAT1 and one of STAT3,
- $S_1 \cdot D_6 \cdot pS_3 =$ dimer bound to one molecule of STAT1 and one of pSTAT3,
- $pS_1 \cdot D_6 \cdot pS_3 =$ dimer bound to one molecule of pSTAT1 and one of pSTAT3,
- $pS_1 =$ unbound cytoplasmic phosphorylated STAT1,
- $pS_3 =$ unbound cytoplasmic phosphorylated STAT3.

The initial reactions in the HypIL-6 signaling pathway can then be described by the ODEs (1) – (22) in the supplementary text, under the law of mass action, where the terms involving the parameter $\beta_6$ apply only to the model under hypothesis 1 and the terms involving the parameter $\gamma_6$ apply only to the model under hypothesis 2. Square brackets around a species denote the concentration of this species with unit nM, and "." implies a reaction bond between two molecules/species. The ODEs are valid for any time $t$, with $t \geq 0$, but time has been omitted in the species concentration for ease of notation. We note here that, for example $[R_1] = [R_1](t)$ for all $t \geq 0$.

Similarly, and with some species in common with the HypIL-6 model, the IL-27 model has been formulated based on biochemical reactions involving the following species:

- $L_{27} =$ IL-27,
- $R_1 =$ GP130,
- $R_2 =$ IL-27Rα,
- $C_2 =$ IL-27Rα - IL-27 monomer,
- $D_{27} =$ phosphorylated IL-27Rα - IL-27 - GP130 heterodimer,
- $S_1 =$ unbound cytoplasmic unphosphorylated STAT1,
- $S_3 =$ unbound cytoplasmic unphosphorylated STAT3,
- $S_1 \cdot D_{27} =$ dimer bound to STAT1 via $R_1$,
- $S_3 \cdot D_{27} =$ dimer bound to STAT3 via $R_1$,
- $pS_1 \cdot D_{27} =$ dimer bound to pSTAT1 via $R_1$,
- $pS_3 \cdot D_{27} =$ dimer bound to pSTAT3 via $R_1$,
- $D_{27} \cdot S_1 =$ dimer bound to STAT1 via $R_2$,
- $D_{27} \cdot S_3 =$ dimer bound to STAT3 via $R_2$,
- $D_{27} \cdot pS_1 =$ dimer bound to pSTAT1 via $R_2$,
- $D_{27} \cdot pS_3 =$ dimer bound to pSTAT3 via $R_2$,
- $S_1 \cdot D_{27} \cdot S_1 =$ dimer bound to two molecules of STAT1,
- $pS_1 \cdot D_{27} \cdot S_1 =$ dimer bound to two molecules of STAT1, phosphorylated on $R_1$,
- $S_1 \cdot D_{27} \cdot pS_1 =$ dimer bound to two molecules of STAT1, phosphorylated on $R_2$,
- $pS_1 \cdot D_{27} \cdot pS_1 =$ dimer bound to two molecules of pSTAT1,
- $S_3 \cdot D_{27} \cdot S_3 =$ dimer bound to two molecules of STAT3,
- $pS_3 \cdot D_{27} \cdot S_3 =$ dimer bound to two molecules of STAT3, phosphorylated on $R_1$,
- $S_3 \cdot D_{27} \cdot pS_3 =$ dimer bound to two molecules of STAT3, phosphorylated on $R_2$,
- $pS_3 \cdot D_{27} \cdot pS_3 =$ dimer bound to two molecules of pSTAT3,
- $S_1 \cdot D_{27} \cdot S_3 =$ dimer bound to STAT1 via $R_1$ and STAT3 via $R_2$,
- $S_3 \cdot D_{27} \cdot S_1 =$ dimer bound to STAT1 via $R_2$ and STAT3 via $R_1$,
- $pS_1 \cdot D_{27} \cdot S_3 =$ dimer bound to pSTAT1 via $R_1$ and STAT3 via $R_2$,
- $S_3 \cdot D_{27} \cdot pS_1 =$ dimer bound to pSTAT1 via $R_2$ and STAT3 via $R_1$,
- $S_1 \cdot D_{27} \cdot pS_3 =$ dimer bound to STAT1 via $R_1$ and pSTAT3 via $R_2$,
- $pS_3 \cdot D_{27} \cdot S_1 =$ dimer bound to STAT1 via $R_2$ and pSTAT3 via $R_1$,
- $pS_1 \cdot D_{27} \cdot pS_3 =$ dimer bound pSTAT1 via $R_1$ and pSTAT3 via $R_2$,
- $pS_3 \cdot D_{27} \cdot pS_1 =$ dimer bound pSTAT3 via $R_1$ and pSTAT1 via $R_1$,
- $pS_1 =$ unbound cytoplasmic phosphorylated STAT1,
- $pS_3 =$ unbound cytoplasmic phosphorylated STAT3.

Again, under the law of mass action, the initial reactions in the IL-27 signaling pathway can be described by the ODEs (23) – (55) in the supplementary text. Similarly, to the HypIL-6 model, the terms in the supplementary Equations (23) - (55) involving the parameter $\beta_{27}$ apply only to the model under hypothesis 1 and the terms involving the parameter $\gamma_{27}$ apply only to the model under hypothesis 2.

We now describe how we have made use of the experimental data (Supp. Fig. 5c and 5d) to parameterize the mathematical models described above. Since the experimental outputs are levels of pSTAT1 and pSTAT3 as a function of time under HypIL-6 and IL-27 stimulation (Supp. Fig. 5c & 5d), we consider two model outputs of interest for the HypIL-6 and IL-27 mathematical models,

which are proportional to the experimental data in Supp. Fig. 5c & 5d; namely, the sum of all molecular species (variables) containing phosphorylated STAT1 (free or bound) ($[pS_1]^{T,j}$, for $j \in \{6, 27\}$) and the sum of all species (variables) containing phosphorylated STAT3 (free or bound) ($[pS_3]^{T,j}$, for $j \in \{6, 27\}$). The total concentrations of the two model outputs of interest at any time $t$ are defined by the following equations, where $T$ denotes the total concentration of the given molecular species:

$$
\begin{aligned}
[pS_1]^{T,6}(t) = & \ [D_6 \cdot pS_1](t) + [pS_1 \cdot D_6 \cdot S_1](t) + 2[pS_1 \cdot D_6 \cdot pS_1](t) + [pS_1 \cdot D_6 \cdot S_3](t) \\
& + [pS_1 \cdot D_6 \cdot pS_3](t) + [pS_1](t)
\end{aligned}
\tag{1}
$$

$$
\begin{aligned}
[pS_3]^{T,6}(t) = & \ [D_6 \cdot pS_3](t) + [pS_3 \cdot D_6 \cdot S_3](t) + 2[pS_3 \cdot D_6 \cdot pS_3](t) + [pS_3 \cdot D_6 \cdot S_1](t) \\
& + [pS_3 \cdot D_6 \cdot pS_1](t) + [pS_3](t)
\end{aligned}
\tag{2}
$$

for the HypIL-6 model, and by

$$
\begin{aligned}
[pS_1]^{T,27}(t) = & \ [pS_1 \cdot D_{27}](t) + [D_{27} \cdot pS_1](t) + [pS_1 \cdot D_{27} \cdot pS_1](t) + [S_1 \cdot D_{27} \cdot pS_1](t) \\
& + 2[pS_1 \cdot D_{27} \cdot pS_1](t) + [pS_1 \cdot D_{27} \cdot S_3](t) + [S_3 \cdot D_{27} \cdot pS_1](t) \\
& + [pS_1 \cdot D_6 \cdot pS_3](t) + [pS_3 \cdot D_6 \cdot pS_1](t) + [pS_1](t),
\end{aligned}
\tag{3}
$$

$$
\begin{aligned}
[pS_3]^{T,27}(t) = & \ [pS_3 \cdot D_{27}](t) + [D_{27} \cdot pS_3](t) + [pS_3 \cdot D_{27} \cdot pS_3](t) + [S_3 \cdot D_{27} \cdot pS_3](t) \\
& + 2[pS_3 \cdot D_{27} \cdot pS_3](t) + [pS_3 \cdot D_{27} \cdot S_1](t) + [S_1 \cdot D_{27} \cdot pS_3](t) \\
& + [pS_1 \cdot D_6 \cdot pS_3](t) + [pS_3 \cdot D_6 \cdot pS_1](t) + [pS_3](t),
\end{aligned}
\tag{4}
$$

for the IL-27 model.

## Bayesian inference

Having developed two mathematical models to describe the experimental system of HypIL-6 and IL-27 stimulation, it was then our objective to parameterize the rates of these models, making use of approximate Bayesian Computation sequential Monte Carlo (ABC-SMC) and Bayesian model selection (selecting between hypothesis 1 and 2, as described in the previous section). In this way, we can learn about which reactions and parameters in the models are regulating the differential signaling by pSTAT1 observed when stimulating with HypIL-6 and IL-27.

Model (hypothesis) selection and parameter estimation are carried out together in the Bayesian sequential Monte Carlo framework used. In the traditional ABC rejection algorithm (*Sunnåker et al., 2013*), the parameters of a mathematical model are sampled from determined prior distributions and the mathematical model is simulated with the sampled (from those prior distributions) parameter values. If the model output is 'close' to the experimental data, as defined by a quantitative measure of distance $d$, that is, if the distance between mathematical model output and experimental data points is smaller than a fiducial threshold $\delta$, that is, $d < \delta$, then the parameters sampled for this instance are accepted into a posterior distribution, otherwise they are rejected. This process is repeated until a posterior of size $N$ is obtained. For ODE models with a small number of parameters and variables, this approach can work well. However, for larger, more complex models, such as the models considered here, this approach can be very time consuming, since one must simulate the model a large number of times in order to obtain a posterior sample of size $N$. This is due to the high dimensionality of the parameter space. For example, in the case of the HypIL-6 model, there are 16 parameters, and thus, a 16-dimensional parameter space. There is, therefore, likely to be large areas of parameter space for which, when the parameters are sampled, the distance measure, $d$, between the model and data is always greater than $\delta$, and hence this procedure is not an efficient sampling strategy.

To this end, we chose to use a more accurate, iterative method, ABC-SMC. At the first iteration, the algorithm works in the same way as the rejection algorithm (described above), where "particles" (parameter sets) are accepted if they result in a distance $d < \delta_1$, where $\delta_1$ is reasonably large; for instance, if approximately 50% of all parameter sets sampled lead to a distance $d < \delta_1$. At each successive iteration, the parameters are sampled not from the prior distributions, but from the posterior distributions which have been obtained from the previous iteration, with weights that depend on

both the prior and the previous iteration distributions. The parameters are then perturbed slightly using a perturbation kernel (for more details see *Toni et al., 2009*) and the model is simulated as above. A particle at iteration $i$ is accepted if it results in a distance measure $d < \delta_i$, where the delta values form a decreasing sequence, $\delta_1 > \delta_2 > \delta_3 > \cdots > \delta_M$, for $M$ iterations. In this way, at each iteration the parameters are sampled from a reduced parameter space region, and hence this approach leads to a faster, more efficient, convergence to the posterior distributions.

A model (hypothesis) selection can be performed making use of the iterative method described above (see *Toni et al., 2009*). We consider a given model (hypothesis) and its parameters are sampled from either the prior (iteration 1) or the previous iterations posterior distributions (iteration $i$ such that $1 < i < M$, in which case the parameters are also perturbed), and again the particle is accepted if the distance between mathematical model output and experimental data points is such that $d < \delta_i$ for iteration $i$. This procedure continues until we obtain a posterior distribution of size $N$ for each mathematical model (hypothesis) considered. The mathematical models are initially given equal weight; that is, they are considered to be equally likely. However, as the iterations of the inference algorithm proceed, if one mathematical model (hypothesis) results in parameter values that better explain the experimental data, this model (hypothesis) will have a greater number of parameter sets accepted per iteration, in the limit $i \to M$. Thus, model selection between two mathematical models (hypotheses) can be determined by the computation of the relative probability for each model (hypothesis) $k$ at iteration $i$, denoted by $P_i^k$, and defined as follows:

$$P_i^k = \frac{\text{number of accepted particles at iteration i from model (hypothesis) k}}{N},$$

for $k \in \{1, 2\}$. The limit $lim_{i \to M} P_i^k$ gives the probability $P^k$ which yields the relative probability of mathematical model $k$ in light of the experimental data.

The data sets we used to compare with the mathematical model outputs were the mean relative fluorescence intensity of total phosphorylated STAT1 and total phosphorylated STAT3 for the cytokine stimulation experiments with both RPE1 and Th-1 cells (Supp. Figure 5b and 5c). Given that the data points have units of mean fluorescence intensity and the model output has dimensions of concentration (in nM units), we normalized the data sets to obtain dimensionless values, which in turn can be compared with the normalized mathematical model outputs. To this end, and firstly, we constructed a linear model for the fluorescence intensity (background fluorescence) of antibodies of phosphorylated STAT1 and STAT3 in unstimulated cells. We then subtracted the value of this linear model at each time point from the corresponding fluorescence intensity in HypIL-6 and IL-27-stimulated cells, for each repeat of the experiment and cell type. Let us denote by $f$ the experimental fluorescence intensity, $f(r, i, tp, j, d)$ corresponds to the fluorescence intensity for the $r$ th repeat, $r \in R = \{1, 2, 3, 4\}$ with antibody for STATi, $i \in I = \{1, 3\}$ at time point $tp$,

$$tp \in TP = \{0\,min, 5\,min, 15\,min, 30\,min, 60\,min, 90\,min, 120\,min, 180\,min\}$$

under stimulation by cytokine IL-j (HypIL-j when $j = 6$), with $j \in J = \{6, 27\}$ and in cell type $d \in D = \{\mathrm{RPE1}, \mathrm{Th}-1\}$. Each data point $data(r, i, tp, j, d)$, to be used in the Bayesian inference and Bayesian model selection was then obtained from $f(r, i, tp, j, d)$ with the following normalization,

$$data(r, i, tp, j, d) = \frac{f(r, i, tp, j, d)}{f(r, i, tp = 30min, j = 27, d)}. \tag{5}$$

That is, our normalization has been chosen to be the time point 30 minutes with IL-27 stimulation. To compare the mathematical model output, denoted by $sim(i, tp, j, d)$, with the normalized experimental data, the mathematical output was normalized in the same way as the data, that is,

$$sim(i, tp, j, d) = \frac{[pS_i]^{T,j}(tp, d)}{[pS_i]^{T,27}(30min, d)}, \tag{6}$$

where $[pS_i]^{T,j}(tp, d)$ denotes the total, $T$, concentration of phosphorylated STATi at time $tp$ (see *Equations 1–4*) when considering cell type $d$ and cytokine stimulation $j \in J = \{6, 27\}$. The data points for IL-27 stimulation at time 30 mins were chosen as normalization in *Equation (5)*, since they correspond to the maximal experimental value, and thus the data, when normalized in this way, was

transformed to a scale in the interval $[0,1]$. Mathematical model outputs were normalized in the same way, and therefore model outputs for IL-27 will naturally go through the value 1 at time 30 mins. This choice of normalization has not influenced the results of our study. To this end, we have repeated our analysis with a different normalization time point of $f(r,i,tp=15min,j=27,d)$ (results not shown here). In particular, we found that the posterior distributions and the result of the model (hypothesis) selection were rather similar for each choice of normalization.

Once data sets and mathematical outputs have been normalized and can be compared, there is a need to quantify how close (or not) they are. To this end, we make use of a quantitative measure, called a distance and denoted by $\delta(sim,data)$. In our case, we have chosen a generalization of the Euclidean distance, where

$$\left[\delta^d(sim,data)\right]^2 = \sum_{i\in I}\sum_{tp\in TP}\sum_{j\in J}\left[sim(i,tp,j,d)-\mu_{data}(i,tp,j,d)\right]^2, \tag{7}$$

$d\in D=\{\mathrm{RPE1,Th-1}\}$, with $\mu_{data}(i,tp,j,d)$ defined as the mean of the four experimental repeats. The mean is given by

$$\mu_{data}(i,tp,j,d) = \frac{1}{4}\sum_{r=1}^{4} data(r,i,tp,j,d). \tag{8}$$

With a choice of measure, we can now carry out both Bayesian (mathematical) model selection and parameter inference. Before we do so, our prior beliefs of the parameters need to be considered and defined. Each of the parameters (reaction rates in the mathematical models) and initial concentrations for the molecules in the model were sampled from a prior distribution, where the distribution was informed by independent experiments or published values, when possible. The choice of prior distributions is given in *Table 3*.

We now provide additional details of the model selection and parameter estimation carried out with the normalized experimental data and mathematical models. We have experimental data for two different cell types (RPE1 and Th-1). There are a number of parameters in the HypIL-6 and IL-27 mathematical models which we have assumed to be independent of the cell type. The only parameters which we would expect to differ between cell types are the initial concentrations of the molecules ($[R_1\ (0)]$, $[R_2\ (0)]$, $[S_1\ (0)]$, $[S_3\ (0)]$), the rate of STAT phosphorylation ($q$), the rates of pSTAT dephosphorylation ($d_1$, $d_3$) and the rates of internalization/degradation of the receptor molecules ($\beta_6$, $\beta_{27}$, $\gamma_6$, $\gamma_{27}$). Therefore, the aforementioned parameters have been estimated for each cell type, separately, and the rest of the parameters have been estimated with one cell type. We have chosen to make use of the RPE1 data to estimate these shared parameters. In particular, we ran the analysis in two stages (all our numerical codes were written in Python), in order to reduce the number of parameters to be estimated in a given ABC-SMC analysis. We first made use of the RPE1 data and secondly, the Th-1 data. We ran the model (hypothesis) selection algorithm with the RPE1 data, for a sample of size $N=10^4$ and $M=15$ iterations, with the following sequence of delta values $\{100,10,5,3,2.5,2.25,2,1.75,1.5,1.25,1.1,1,0.9,0.8,0.7\}$.

A uniform perturbation kernel (*Filippi et al., 2013*) was used to perturb the parameters sampled at each iteration $1<i\le M$. The result of the model selection clearly indicated that hypothesis 1 could explain the experimental data best.

When carrying out model (hypothesis) selection, and during the latter iterations, there were still some parameter sets accepted for hypothesis 2. Therefore, we ran the ABC-SMC parameter estimation algorithm for the RPE1 data only for hypothesis 1, in order to generate a complete posterior distribution of size $N=10^4$. In the second stage of the inference, we then considered the Th-1 data and ran the ABC-SMC parameter estimation algorithm to infer Th-1 specific parameters (only for hypothesis 1). Those parameters that were assumed to be independent of cell type were sampled from the posterior distributions obtained during the previous RPE1 ABC-SMC inference. The same sample size, number of iterations, vector of delta values and perturbation kernel were used in the Th-1 and RPE1 analysis. We finally note that, although we used the experimental data sets in the order RPE1 followed by Th-1 here, the ordering of the cell type does not have a significant effect on the results (not shown here). Namely, our model selection results and those of the parameter inference and posterior distributions are similar to those presented here, when we reverse the order.

**Table 3.** Prior distribution assigned to each parameter and initial molecular concentration.
*These distributions are centered around measurements obtained from cell surface receptor quantification experiments. **These distributions were derived based on $K_d$ values obtained from the literature (*Wiederkehr-Adam et al., 2003*). ***These distributions are based on values derived from experimental data in which the cells were treated with Tofacitinib. † These distributions were based on values derived from experimental data in which the cells were treated with cycloheximide. ‡These distributions were based on computations involving approximate cell sizes and average numbers of molecules per cell.

| Parameter | Prior distribution | Reference |
|---|---|---|
| $r_{1,6}^+$ | $10^r$ for $r \sim N(-3, 1.5)$ | * |
| $r_{1,6}^-$ | $10^r$ for $r \sim N(-3.9, 1.96)$ | * |
| $r_{1,27}^+$ | $10^r$ for $r \sim N(-2.34, 1.17)$ | * |
| $r_{1,27}^-$ | $10^r$ for $r \sim N(-2.82, 1.41)$ | * |
| $r_{2,j}^+$ for $j \in \{6, 27\}$ | $10^r$ for $r \sim Unif(-2, 3)$ | *Kozer et al., 2013* |
| $r_{2,j}^-$ for $j \in \{6, 27\}$ | $10^r$ for $r \sim Unif(-3, 1)$ | *Kozer et al., 2013* |
| $k_{ia}^+, k_{ib}^+$ for $i \in \{1, 3\}$ | $10^r$ for $r \sim Unif(-7, 1)$ | ** |
| $k_{ia}^-, k_{ib}^-$ for $i \in \{1, 3\}$ | $10^r$ for $r \sim Unif(-2, 1)$ | ** |
| $q$ | $10^r$ for $r \sim Unif(-3, 2)$ | Assumed |
| $d_i$ for $i \in \{1, 3\}$ | $10^r$ for $r \sim Unif(-5, -2)$ | *** |
| $\beta_j$ for $j \in \{6, 27\}$ | $10^r$ for $r \sim Unif(-5, -1)$ | † |
| $\gamma_j$ for $j \in \{6, 27\}$ | $10^r$ for $r \sim Unif(-5, -1)$ | † |
| $[R_1(0)]$ | $N(12.7, 6.35)$ | ‡ |
| $[R_2(0)]$ | $N(33.8, 16.9)$ | ‡ |
| $[S_1(0)]$ | $N(300, 100)$ | *Itzhak et al., 2016* |
| $[S_3(0)]$ | $N(400, 100)$ | *Itzhak et al., 2016* |

## Mathematical model validation

Once we have carried out model selection and parameter inference as described in the previous section, we want to explore the potential of the mathematical models of cytokine early signaling to reproduce other independent experiments. We note that we make use of the posterior parameter distributions obtained from the RPE1 data ABC-SMC. Our first interest was to reproduce the dose response curve seen in Supp. Fig. 2a. To this end, we run both models using the $10^4$ accepted parameters sets from the ABC-SMC for 18 different values of cytokine concentration, within the range $[10^{-4}\text{-}10^2]$ log nM. The results of this analysis are seen in Supp. Fig. 13b. Our second interest was to describe the IL-27Rα-GP130 chimera experiments (*Figure 3c*). This required us to modify the mathematical model as follows: we considered the ODEs of the IL-27 mathematical model involved in the formation of the dimer, Equations (23) – (26) (Supplementary information 1), and the ODEs of the HypIL-6 mathematical model post-dimer formation, Equations (5) – (22) (Supplementary information 1), in which $D_6$ was replaced by $D_{27}$. The ODE of the IL-27 induced dimer in the chimera model was modified as follows,

$$\begin{aligned}\frac{d[D_{27}]}{dt} &= r_{2,27}^+[C_2][R_1] - r_{2,27}^-[D_{27}] - 2k_{1a}^+[D_{27}][S_1] + k_{1a}^-([S_1^\cdot D_{27}] + [pS_1^\cdot D_{27}]) \\ &\quad - 2k_{3a}^+[D_{27}][S_3] + k_{3a}^-([S_3^\cdot D_{27}] + [pS_3^\cdot D_{27}]) - \beta_{27}[D_{27}].\end{aligned} \tag{9}$$

We simulated both the original mathematical model of IL-27 and the chimera model using the accepted parameter sets from the ABC-SMC. The results can be seen in Supp. Fig. 3a.

Finally, the mathematical model was used to describe one of the mutant varieties of the IL-27Rα chain, Y613F. We sought to reproduce the results of *Figure 3b* making use of the mathematical model of IL-27 signaling. Since the Y613F mutation decreases the affinity of STAT1 to IL-27Rα, we fixed the association and dissociation rates of STAT1 to the IL 27Rα chain, $k_{1b}^+$ and $k_{1b}^-$, respectively,

at values which lead to a high affinity (of the order of µM values). The specific values chosen were $k_{1b}^+ = 10^{-5}$ nM$^{-1}$s$^{-1}$ and $k_{1b}^- = 10^1$ s$^{-1}$, which yield an affinity of $10^2$µM. The rate $k_{1b}^-$, was chosen as approximately the median of the ABC-SMC posterior distribution for this parameter, and the rate $k_{1b}^+$ was then significantly decreased in order to increase the affinity. We simulated the mathematical model of IL-27 signaling using the $10^4$ accepted ABC-SMC parameter sets, but where the rates $k_{1b}^+$ and $k_{1b}^-$, were fixed as described above. The pointwise medians and 95% credible intervals of these simulations are plotted in Supp. Fig. 13c, as well as the simulations for the WT IL-27Rα chain, without altering any of the parameter values from the posterior distributions. Altering the binding affinity of STAT1 to IL-27Rα in this way in the mathematical model allows us to generate results which qualitatively and quantitatively reproduce the experimental observations for the Y613F mutant from *Figure 3b*.

## Live-cell dual-color single-molecule imaging studies

Single-molecule imaging experiments were carried out by total internal reflection fluorescence (TIRF) microscopy with an inverted microscope (Olympus IX71) equipped with a triple-line total internal reflection (TIR) illumination condenser (Olympus) and a back-illuminated electron multiplied (EM) CCD camera (iXon DU897D, 512 × 512 pixel, Andor Technology) as recently described (*Wilmes et al., 2015*; *Wilmes et al., 2020*; *Moraga et al., 2015b*). A ×150 magnification objective with a numerical aperture of 1.45 (UAPO 150 3/1.45 TIRFM, Olympus) was used for TIR illumination. All experiments were carried out at room temperature in medium without phenol red supplemented with an oxygen scavenger and a redox-active photoprotectant to minimize photobleaching (*Vogelsang et al., 2008*). For Heterodimerization experiments of IL-27Rα and GP130 cell surface labeling of RPE1 GP130 KO, co-transfected with mXFPe-IL-27Rα and mXFPm-GP130, was achieved by adding αGFP-enNB$^{RHO11}$ and αGFP-miNB$^{DY647}$ to the medium at equal concentrations (5 nM) and incubated for at least 5 min prior to stimulation with IL-27 (20 nM) or HypIL-6 (20 nM). For homodimerization experiments with mXFPm-GP130, αGFP-miNB$^{DY647}$ and αGFP-miNB$^{RHO11}$ (*Kirchhofer et al., 2010*) were used for cell surface receptor labeling as described above. The nanobodies were kept in the bulk solution during the whole experiment in order to ensure high equilibrium binding to mXFP-GP130. For simultaneous dual-color acquisition, αGFP-NB$^{RHO11}$ was excited by a 561 nm diode-pumped solid-state laser at 0.95 mW (~32 W/cm$^2$) and αGFP-NB$^{DY647}$ by a 642 nm laser diode at 0.65 mW (~22 W/cm$^2$). Fluorescence was detected using a spectral image splitter (DualView, Optical Insight) with a 640 DCXR dichroic beam splitter (Chroma) in combination with the bandpass filter 585/40 (Semrock) for detection of RHO11 and 690/70 (Chroma) for detection of DY647 dividing each emission channel into 512 × 256 pixel. Image stacks of 150 frames were recorded at 32 ms/frame.

Single-molecule localization and single-molecule tracking were carried out using the multiple-target tracing (MTT) algorithm (*Sergé et al., 2008*) as described previously (*You et al., 2016*). Step-length histograms were obtained from single-molecule trajectories and fitted by two fraction mixture model of Brownian diffusion. Average diffusion constants were determined from the slope (2–10 steps) of the mean square displacement versus time lapse diagrams. Immobile molecules were identified by the density-based spatial clustering of applications with noise (DBSCAN) algorithm as described recently (*Röder et al., 2014*). For comparing diffusion properties and for co-tracking analysis, immobile particles were excluded from the data set. Prior to co-localization analysis, imaging channels were aligned with sub-pixel precision by using a spatial transformation. To this end, a transformation matrix was calculated based on a calibration measurement with multicolor fluorescent beads (TetraSpeck microspheres 0.1 mm, Invitrogen) visible in both spectral channels (cp2tform of type 'affine', The MathWorks MATLAB 2009a).

Individual molecules detected in the both spectral channels were regarded as co-localized, if a particle was detected in both channels of a single frame within a distance threshold of 100 nm radius. For single-molecule co-tracking analysis, the MTT algorithm was applied to this dataset of co-localized molecules to reconstruct co-locomotion trajectories (co-trajectories) from the identified population of co-localizations. For the co-tracking analysis, only trajectories with a minimum of 10 steps (~320 ms) were considered in order to robustly remove random receptor co-localizations (*Wilmes et al., 2020*). For heterodimerization experiments of mXFPe-IL-27Rα and mXFPm-GP130, the relative fraction of dimerized receptors was calculated from the number of co-trajectories

relative to the number of IL-27Rα trajectories. GP130 was expressed in moderate excess (~1.5–2 fold), so that maximal receptor assembly was not limited by abundance of the low-affinity subunit GP130.

For homodimerization experiments with GP130, the relative fraction of co-tracked molecules was determined with respect to the absolute number of trajectories and corrected for GP130 stochastically double-labeled with the same fluorophore species as follows:

$$AB^* = \frac{AB}{2 \times \left[ \left( \frac{A}{A+B} \right) \times \left( \frac{B}{A+B} \right) \right]}, rel.co - locomotion = \frac{2 \times AB^*}{(A+B)}$$

where A, B, AB, and AB* are the numbers of trajectories observed for Rho11, DY647, co-trajectories and corrected co-trajectories, respectively.

The two-dimensional equilibrium dissociation constants ($K_D^{2D}$) were calculated according to the law of mass action for a monomer-dimer equilibrium:

Heterodimerization (IL-27Rα + GP130):

$$K_D^{2D} = \frac{([GP130] - (\alpha \times [IL27Ra])) \times ([IL27Ra] - (\alpha \times [IL27Ra]))}{(\alpha \times [IL27Ra])}$$

or

$$K_D^{2D} = [GP130] \times \left( \frac{1}{\alpha} - 1 \right) + [IL27Ra] \times (\alpha - 1)$$

with:

$$\alpha = fraction\,of\,IL27\,bound\,IL27R\alpha\,in\,complex\,with\,GP130$$

Homodimerization (GP130 + GP130):

$$K_D^{2D} = \frac{[M]^2}{[D]} = \frac{\left( [M]_0 - 2[D] \right)^2}{[D]}$$

$$K_D^{2D} = \frac{([GP130] - 2 \times (\alpha \times [GP130]))^2}{2 \times (\alpha \times [GP130])}$$

with:

$$\alpha = fraction\,of\,GP130\,homodimers\,relative\,to\,[GP130]/2$$

where [M] and [D] are the concentrations of the monomer and the dimer, respectively, and $[M]_0$ is the total receptor concentration.

## Acknowledgements

We thank members of the Moraga, Molina-París, Piehler and Mitra laboratories for helpful advice and discussion. We thank G Hikade and H Kenneweg for technical support, C P Richter for providing software for single-molecule image analysis, R Kurre (Integrated Bioimaging Facility Osnabrück) for support with fluorescence microscopy and the FingerPrints Proteomics facility (Dundee) for support with the mass spectrometry data. Graphic material was provided by smart servier medical art. This work was supported by the Wellcome-Trust-202323/Z/16/Z (IM EP), ERC-206-STG grant (IM JMF EP PKF), EMBO (SW 454–2017), DFG (SFB 944, P8/Z, JP), National Heart, Lung and Blood Institute (K22HL125593, MK) and Contrat de Plan Etat Région Hauts de France and Institut pour la Recherche sur le Cancer de Lille (SM SG). CMP and GL were supported by H2020 (ITN QuanTII, grant agreement number 764698). PJ is supported by the EPSRC, AstraZeneca and the Smith Institute (Smith Institute CASE studentship, award reference 1969354). Numerical work was undertaken on ARC3, which is part of the High-Performance Computing facilities at the University of Leeds, UK.

## Additional information

### Funding

| Funder | Grant reference number | Author |
|---|---|---|
| Horizon 2020 Framework Programme | 714680 | Stephan Wilmes<br>Jonathan Martinez-Fabregas<br>Paul K Fyfe<br>Ignacio Moraga |
| Wellcome Trust | 202323/Z/16/Z | Elizabeth Pohler<br>Ignacio Moraga |
| EPSRC | 1969354 | Polly-Anne Jeffrey |
| University of Leeds | 764698 | Grant Lythe<br>Carmen Molina-París |
| Wellcome Trust | ERC-206-STG | Jonathan Martinez-Fabregas<br>Paul K Fyfe<br>Elizabeth Pohler<br>Ignacio Moraga |
| EMBO | 454–2017 | Stephan Wilmes |
| DFG | SFB 944, P8/Z | Jacob Piehler |
| National Heart, Lung, and Blood Institute | K22HL125593 | Jacob Piehler |

The funders had no role in study design, data collection and interpretation, or the decision to submit the work for publication.

### Author contributions

Stephan Wilmes, Conceptualization, Data curation, Formal analysis, Investigation, Methodology, Writing - original draft; Polly-Anne Jeffrey, Conceptualization, Data curation, Formal analysis, Methodology, Writing - review and editing; Jonathan Martinez-Fabregas, David Launay, Investigation, Methodology, Writing - review and editing; Maximillian Hafer, Paul K Fyfe, Elizabeth Pohler, Silvia Gaggero, Suman Mitra, Conceptualization, Investigation, Methodology, Writing - review and editing; Martín López-García, Formal analysis, Methodology; Grant Lythe, Conceptualization, Writing - review and editing; Charles Taylor, Formal analysis; Thomas Guerrier, Methodology; Jacob Piehler, Conceptualization, Methodology, Writing - review and editing; Carmen Molina-París, Conceptualization, Supervision, Investigation, Methodology, Writing - review and editing; Ignacio Moraga, Conceptualization, Data curation, Supervision, Funding acquisition, Investigation, Writing - original draft, Project administration, Writing - review and editing

### Author ORCIDs

Stephan Wilmes (iD) https://orcid.org/0000-0002-4112-710X
Polly-Anne Jeffrey (iD) https://orcid.org/0000-0001-6476-0402
Jonathan Martinez-Fabregas (iD) http://orcid.org/0000-0001-5809-065X
Maximillian Hafer (iD) http://orcid.org/0000-0003-0853-2637
Paul K Fyfe (iD) http://orcid.org/0000-0003-3541-2294
Martín López-García (iD) https://orcid.org/0000-0003-3833-8595
Grant Lythe (iD) https://orcid.org/0000-0001-7966-5571
Jacob Piehler (iD) http://orcid.org/0000-0002-2143-2270
Ignacio Moraga (iD) https://orcid.org/0000-0001-9909-0701

### Ethics

Human subjects: This study was authorized by the French Competent Authority dealing with Research on Human Biological Samples namely the French Ministry of Research. The Authorization number is ECH 19/04. all patients gave their written informed consent.

Decision letter and Author response
Decision letter https://doi.org/10.7554/eLife.66014.sa1
Author response https://doi.org/10.7554/eLife.66014.sa2

## Additional files

### Supplementary files

• Supplementary file 1. Ordinary differential equations for mathematical modeling of IL-27 and HypIL-6 signaling.

• Transparent reporting form

### Data availability

Python (version 3.7) codes for the ABC-SMC model selection and parameter inference can be found in the public repository 'https://github.com/PollyJeffrey/Cytokine_modelling' (copy archived at https://archive.softwareheritage.org/swh:1:rev:9c3e0ddc7a96eac941baad560d1541d660b0515d), along with the results of the analysis. Phospho-proteomic and proteomic datasets were uploaded to the Proteome Exchange platform with accession numbers PXD024657 and PXD024188 respectively. RNA-seq dataset was uploaded in the GSE database with accession number GSE164479.

The following datasets were generated:

| Author(s) | Year | Dataset title | Dataset URL | Database and Identifier |
|---|---|---|---|---|
| Wilmes S, Jeffrey P, Martinez-Fabregas J, Hafer M, Fyfe P, Pohler E, Gaggero S, López-García M, Lythe G, Guerrier T, Launay D, Suman M, Piehler J, Molina-París C, Moraga I | 2021 | Competitive binding of STATs to receptor phospho-Tyr motifs accounts for altered cytokine responses in autoimmune disorders | https://www.ncbi.nlm.nih.gov/geo/query/acc.cgi?acc=GSE164479 | NCBI Gene Expression Omnibus, GSE164479 |
| Wilmes S, Jeffrey P, Martinez-Fabregas J, Hafer M, Fyfe P, Pohler E, Gaggero S, López-García M, Lythe G, Guerrier T, Launay D, Suman M, Piehler J, Molina-París C, Moraga I | 2021 | Competitive binding of STATs to receptor phospho-Tyr motifs accounts for altered cytokine responses in autoimmune disorders | https://www.ebi.ac.uk/pride/archive/projects/PXD024188 | PRIDE, PXD024188 |
| Wilmes S, Jeffrey P, Martinez-Fabregas J, Hafer M, Fyfe P, Pohler E, Gaggero S, López-García M, Lythe G, Guerrier T, Launay D, Suman M, Piehler J, Molina-París C, Moraga I | 2021 | Competitive binding of STATs to receptor phospho-Tyr motifs accounts for altered cytokine responses in autoimmune disorders | https://www.ebi.ac.uk/pride/archive/projects/PXD024657 | PRIDE, PXD024657 |

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
