## [Decision Letter]

**Acceptance summary:**

This study is a great example of an elaborated combination of experimental and mathematical analyses to examine an intriguing, pleiotropic immunological signaling pathway. While a good number of individual aspects of this signaling pathway have been studied and reported before, the presented work puzzles together many pieces and succeeds to present a conclusive and comprehensive model of this particular cytokine system. The main conclusions are well supported by the presented data and the manuscript will be of interest and relevance for the study of many other cytokine signaling pathways, being of broad relevance for immunologists and cell biologists.

**Decision letter after peer review:**

Thank you for submitting your article "Competitive binding of STATs to receptor phospho-Tyr motifs accounts for altered cytokine responses in autoimmunity" for consideration by *eLife*. Your article has been reviewed by 3 peer reviewers, one of whom is a member of our Board of Reviewing Editors, and the evaluation has been overseen by Jos van der Meer as the Senior Editor. The following individual involved in review of your submission has agreed to reveal their identity: Kevin Thurley (Reviewer #2).

Essential Revisions:

All reviewers appreciated the elaborated and thorough analyses presented in this paper. Given this exhaustive amount of work leading to this comprehensive analyses, all reviewers also noticed that the paper is very complex and extensive. To help the reader focus, they recommend to re-think if some material and text could not be shortened or shifted to the discussion or supplement.

The detailed comments below address some specific points that the reviewers noted. In particular this addresses the following main points:

1. Please reconsider the title of the manuscript, as focusing on autoimmunity does not seem to fully capture your work

2. Some aspects regarding the statistical analyses and the mathematical modeling (e.g. hypotheses considerations) need to be explained in more detail and clarified.

3. Please consider to show a full characterization of the RPE1-cells that were used (see comment Reviewer 3)

Reviewer #1 (Recommendations for authors):

– Combining the mathematical model with the experimental data, a Bayesian Model selection and parameter inference approach is used to select between two model hypothesis and, later on, perform parameter estimation. It is not totally clear to me how an upfront Bayesian model selection can be done without addressing the issue of parameter inference. Therefore, shouldn't the comparison of the two model hypotheses not also been applied to the Th1-cell dataset, and not only the RPE1?

– (Figure 2c): The measurement of IL-27 at 30 minutes for REP1 and Th1 does not seem to contain any statistical variation. This also seems to force the model to match this data point for any parameter combination. What is the reason that there is no variation within this data point? As this also seems to be a critical point for differences between the dynamics caused by the two cytokines, would model predictions change if e.g. random variation is added to this data point?

– It is indicated in the text that the parameter inference by approximate Bayesian computation revealed statistically significant differences between the binding rates of certain receptors/molecular (p.7 top). Could you comment on the statistical tests used to compare the posterior distributions of the respective parameters to make this claim? The posterior distributions nicely show a deviation of the parameter estimates from the uniform prior distributions. However, especially in case of multi-parametric and high-dimensional models as here, there can still be the problem of correlating individual parameter estimates inhibiting parameter identifiability. To which extent are individual parameter estimates correlated within the accepted parameter combinations? In addition, more details on the ABC-approach should be given algorithm/software used, required number of particles accepted, possible number of runs (i.e., updating prior distribution by found posterior).

*Reviewer #3 (Recommendations for authors):*I commend the authors for an exhaustive amount of work on this exciting signaling pathway. I would like to focus on a few points now that I have a slightly more critical view on.1. Most individual findings have been reported before (or at least there have been indications). Understandably, this makes defining a clear punch line difficult, but honestly, focussing on "autoimmune diseases" already in the title to me appears a bit farfetched. The SLE aspect is only touched upon in the manuscript, and it is neither very informative nor central to the story. It is good to have it in, just to illustrate natural conditions in which altered STAT levels shapes cytokine responses, but it clearly should not be the selling point.

2. An aspect that I furthermore miss in the introduction and discussion part is the fact, that basically all cytokine receptor systems appear to have a rather strong preference for certain STATs. It appears rather trivial that different receptors / receptor chains will have different affinities for the different STATs and this has been addressed in a few studies previously (e.g. STAT1, STAT3 in ref. 42), or STAT2 in IFNAR1 in PMID:8605876. The present manuscript arrived at the conclusion of differing affinities of STAT1 and STAT3 for GP130 and IL-27Ra by rather intricate modeling approaches and confirms this biochemically. This is OK, but I find it a bit disconcerting that this is sold as a fascinating new finding, while I would say it is the most expected explanation. I dint want to diminish this achievement, just strongly suggest toning this down a bit and, most importantly, add a few words on the known and suspected differential affinities of the various STATs to cytokine receptors in general.

3. In Figure 1d, GP130 was over expressed in RPE1 cells and the authors' conclusion was that GP130 levels do not contribute to sustained pSTAT1. In fact, Figure 1d does look as if pSTAT1 *is* sustained; particularly when you compared to the normalized plot of Th1 cells in Suppl Figure 1c, where the authors mark this difference even in red with δ signs. It would be very interesting to know if 10xGP130 cells stimulated with (Typ)IL-6 would now also show enhanced expression of IRF1 and a second wave of transcription?

4. I would recommend the authors to show a full characterization of the RPE1 cells they use (e.g. for model parameterization and most further experiments) analogously to Th1 cells in Figure 1b and c (i.e. in terms of IL-27 vs IL-6 signaling) in one comprehensive supplementary figure. Suppl. Figure 4 goes in that direction, but it would be good to have this in the beginning (before the modeling) and in an identical fashion to the Th1 experiment. Please further make sure to identify more clearly (in text and/or legends or even figure labels) which cells were used in which experiment (e.g. "RPE1-IL-27Ra" or "RPE1-IL-27-Ra/GP130").

5. In the modeling figure, Figure 2c, I assume the Y-axes labels were swapped? If not, this needs to be explained!

---

## [Author Response]

Essential Revisions:All reviewers appreciated the elaborated and thorough analyses presented in this paper. Given this exhaustive amount of work leading to this comprehensive analyses, all reviewers also noticed that the paper is very complex and extensive. To help the reader focus, they recommend to re-think if some material and text could not be shortened or shifted to the discussion or supplement.The detailed comments below address some specific points that the reviewers noted. In particular this addresses the following main points:1. Please reconsider the title of the manuscript, as focusing on autoimmunity does not seem to fully capture your work2. Some aspects regarding the statistical analyses and the mathematical modeling (e.g. hypotheses considerations) need to be explained in more detail and clarified.3. Please consider to show a full characterization of the RPE1-cells that were used (see comment Reviewer 3)

We thank the reviewers for their very positive comments. Following their advice, we have modified the title of the manuscript, provided a more detailed description of the mathematical model and consolidated all data characterizing RPE1 cells to the same Supplementary Figure 2. Additionally, we have moved Figure 6 to Supp. Figure 11 to streamline the text and simplify the manuscript. Please find below our answers to the specific comments raised by the reviewers.

Reviewer #1 (Recommendations for authors):– Combining the mathematical model with the experimental data, a Bayesian Model selection and parameter inference approach is used to select between two model hypothesis and, later on, perform parameter estimation. It is not totally clear to me how an upfront Bayesian model selection can be done without addressing the issue of parameter inference. Therefore, shouldn't the comparison of the two model hypotheses not also been applied to the Th1-cell dataset, and not only the RPE1?

As the reviewer implies, mathematical model selection and parameter estimation are carried out in parallel. We have clarified this issue and the details of the analysis in the Materials and methods, Bayesian inference section. We chose to carry out the Bayesian inference sequentially, i.e., in RPE1 cells followed by Th-1 cells in order to reduce the total number of parameters to be inferred during any one analysis (since a subset of the parameters were shared between cell types). The model selection was therefore carried out in the RPE1 cells first, and the result of this inference was carried on to the Th-1 cell analysis. Hence the decision only to carry out parameter estimation for hypothesis 1 in the Th-1 cells. The ordering of the cell types in the analysis was chosen since we had better initial estimates of receptor and STAT concentrations in RPE1 cells. We have however carried out the analysis in the opposite order as well, Th-1 followed by RPE1, and the result of the model selection and parameter inference does not change.

– (Figure 2c): The measurement of IL-27 at 30 minutes for REP1 and Th1 does not seem to contain any statistical variation. This also seems to force the model to match this data point for any parameter combination. What is the reason that there is no variation within this data point? As this also seems to be a critical point for differences between the dynamics caused by the two cytokines, would model predictions change if e.g. random variation is added to this data point?

The reviewer is correct that there is no variation at this point. This is due to the normalisation of the experimental data, as in Equation 5. The mathematical model output was normalised in the same way (Equation 6) so that the model naturally passed through this point. The normalisation was required since the dimensions of the data and model output were not the same. The normalisation point of IL-27 at time 30 minutes was chosen since it is the maximum value in the data set, and hence after normalisation the data was transformed to a range of [0; 1]. This normalisation has been clarified in the Materials and methods, Bayesian inference section. In order to check that the choice of normalisation point did not have a dramatic effect on the results of the analysis, we carried out the model selection and parameter estimation using a different normalisation point (IL-27 at time 15 minutes), and the results were rather similar.

– It is indicated in the text that the parameter inference by approximate Bayesian computation revealed statistically significant differences between the binding rates of certain receptors/molecular (p.7 top). Could you comment on the statistical tests used to compare the posterior distributions of the respective parameters to make this claim? The posterior distributions nicely show a deviation of the parameter estimates from the uniform prior distributions. However, especially in case of multi-parametric and high-dimensional models as here, there can still be the problem of correlating individual parameter estimates inhibiting parameter identifiability. To which extent are individual parameter estimates correlated within the accepted parameter combinations? In addition, more details on the ABC-approach should be given algorithm/software used, required number of particles accepted, possible number of runs (i.e., updating prior distribution by found posterior).

We have carried out a structural identifiability analysis for both mathematical models, and the result of this analysis indicated that all parameters were individually structurally identifiable. We have added a comment about this analysis in the main text. Whilst it is true that there are some strong correlations between individual pairs of parameters in their posterior distributions, the particular parameters that are discussed in the main text (i.e., the pairs of STAT/receptor interaction parameters on the second row of Figure 2d) are not strongly correlated and thus, we can be confident that the posterior distributions are representative of the rates we wished to infer. We have added a table of summary statistics for the posterior distributions of these parameters, to further elucidate the differences found. We also carried out two-sample paired t-tests for the means of the distributions for each pair of STAT/receptor parameters, after thinning of the distributions (taking every 100th value from each posterior distribution, resulting in thinned distributions with size 10^2^). The thinning was necessary since the sample size was so large (N = 10^4^) that the t-test would reveal a significant difference between any pair of parameters considered in Figure 2d. We found when using a thinned sample of size 10^2^, that there were significant differences between the means of the STAT/receptor parameter pairs at the 5% level, as expected. To clarify that these significant differences were not still due to the relatively large sample size (10^2^), we carried out t-tests using the same method for the parameter pairs on the top row of Figure 2d (ligand binding and dimerisation parameters) and found that the means of these pairs of distributions were not significantly different at the 5% level. We have not included the t-test analysis in the manuscript. Finally, we have added to the Materials and methods, Bayesian inference section, more details about the ABC approach used as suggested by the reviewer. We have also provided a link to a GitHub repository which contains the Python codes of the main analyses carried out here (Bayesian model selection and parameter inference). The results of these analyses are also included in the GitHub repository.

Reviewer #3 (Recommendations for authors):I commend the authors for an exhaustive amount of work on this exciting signaling pathway. I would like to focus on a few points now that I have a slightly more critical view on.1. Most individual findings have been reported before (or at least there have been indications). Understandably, this makes defining a clear punch line difficult, but honestly, focussing on "autoimmune diseases" already in the title to me appears a bit far fetched. The SLE aspect is only touched upon in the manuscript, and it is neither very informative nor central to the story. It is good to have it in, just to illustrate natural conditions in which altered STAT levels shapes cytokine responses, but it clearly should not be the selling point.

We thank the reviewer for his comment. We agree with the reviewer that this part is not the most integral part of the study and therefore we have changed the title of our manuscript to reflect this.

2. An aspect that I furthermore miss in the introduction and discussion part is the fact, that basically all cytokine receptor systems appear to have a rather strong preference for certain STATs. It appears rather trivial that different receptors / receptor chains will have different affinities for the different STATs and this has been addressed in a few studies previously (e.g. STAT1, STAT3 in ref. 42), or STAT2 in IFNAR1 in PMID:8605876. The present manuscript arrived at the conclusion of differing affinities of STAT1 and STAT3 for GP130 and IL-27Ra by rather intricate modeling approaches and confirms this biochemically. This is OK, but I find it a bit disconcerting that this is sold as a fascinating new finding, while I would say it is the most expected explanation. I dint want to diminish this achievement, just strongly suggest toning this down a bit and, most importantly, add a few words on the known and suspected differential affinities of the various STATs to cytokine receptors in general.

We agree with the reviewer in that the observation that cytokine receptors all have a certain preference for STAT activation has been well described in literature. However, in recent years the cytokine field has started to realize that cytokine signalling signatures are very plastic and cytokine responses can change dramatically depending on the environment where the responsive cells is localized. This ultimately contributes to the large functional pleiotropy exhibited by cytokines. Our work now provides molecular bases to support these initial observations. We show that the STAT-receptor preference is not as strong as initially believed and that small alterations in STAT and receptor concentration, as those found in pro-inflammatory environments (e.g. SLE) can dramatically change the response of a cell to a given cytokine. In addition, we show that IL-27 has evolved away from this general mechanism of cytokine signalling. By decoupling the binding of STAT1 and STAT3 to IL-27Ra and GP130 respectively, the IL-27 system shows unique properties that allows it to cope with the changing environments. To follow the reviewer suggestions, we have added a paragraph to the introduction giving some background on different STAT affinities to the receptor phospho-Tyrosine motifs and how this would lead to a STAT competition model.

3. In Figure 1d, GP130 was over expressed in RPE1 cells and the authors' conclusion was that GP130 levels do not contribute to sustained pSTAT1. In fact, Figure 1d does look as if pSTAT1 is sustained; particularly when you compared to the normalized plot of Th1 cells in Suppl Figure 1c, where the authors mark this difference even in red with δ signs. It would be very interesting to know if 10xGP130 cells stimulated with (Typ)IL-6 would now also show enhanced expression of IRF1 and a second wave of transcription?

We thank the reviewer for this suggestion. We have performed additional experiments comparing “RPE1 IL27Ra” and “RPE1 10x[GP130]” in their ability to induce IRF1 protein in response to IL-27 or HypIL-6 stimulation (Author response image 1). Interestingly, we found that IRF1 levels are equally low in the course of HypIL-6 stimulation regardless how much GP130 was present. Due to the large volume of data that the manuscript already has, we have decided to not include this data.

4. I would recommend the authors to show a full characterization of the RPE1 cells they use (e.g. for model parameterization and most further experiments) analogously to Th1 cells in Figure 1b and c (i.e. in terms of IL-27 vs IL-6 signaling) in one comprehensive supplementary figure. Suppl. Figure 4 goes in that direction, but it would be good to have this in the beginning (before the modeling) and in an identical fashion to the Th1 experiment. Please further make sure to identify more clearly (in text and/or legends or even figure labels) which cells were used in which experiment (e.g. "RPE1-IL-27Ra" or "RPE1-IL-27-Ra/GP130").

We agree that having a full characterization of the “RPE1 IL-27Ra” model system prior to the modelling part will clearly help to understand the outcome of the modelling approach. Therefore, we have rearranged the supp. figures and now provide a characterization of that model system in Supp. Figure 2. Further, we have added labels indicating which cell system was used in certain experiments (i.e. in supp. Figure 11).

5. In the modeling figure, Figure 2c, I assume the Y-axes labels were swapped? If not, this needs to be explained!

We have corrected this error.